# Causal integration of multi-omics data with prior knowledge to generate mechanistic hypotheses

Aurelien Dugourd[1,2,3,4,†] (iD), Christoph Kuppe[3,4,5,†] (iD), Marco Sciacovelli[6,†], Enio Gjerga[1,2] (iD), Attila Gabor[1], Kristina B. Emdal[7], Vitor Vieira[8], Dorte B. Bekker-Jensen[7], Jennifer Kranz[3,9,10], Eric.M.J. Bindels[11], Ana S.H. Costa[6,‡], Abel Sousa[12,13], Pedro Beltrao[13] (iD), Miguel Rocha[8], Jesper V. Olsen[7] (iD), Christian Frezza[6,*] (iD), Rafael Kramann[3,4,5,**] (iD) & Julio Saez-Rodriguez[1,2,14,***] (iD)

## Abstract

Multi-omics datasets can provide molecular insights beyond the sum of individual omics. Various tools have been recently developed to integrate such datasets, but there are limited strategies to systematically extract mechanistic hypotheses from them. Here, we present COSMOS (Causal Oriented Search of Multi-Omics Space), a method that integrates phosphoproteomics, transcriptomics, and metabolomics datasets. COSMOS combines extensive prior knowledge of signaling, metabolic, and gene regulatory networks with computational methods to estimate activities of transcription factors and kinases as well as network-level causal reasoning. COSMOS provides mechanistic hypotheses for experimental observations across multi-omics datasets. We applied COSMOS to a dataset comprising transcriptomics, phosphoproteomics, and metabolomics data from healthy and cancerous tissue from eleven clear cell renal cell carcinoma (ccRCC) patients. COSMOS was able to capture relevant crosstalks within and between multiple omics layers, such as known ccRCC drug targets. We expect that our freely available method will be broadly useful to extract mechanistic insights from multi-omics studies.

**Keywords** causal reasoning; kidney cancer; metabolism; multi-omics; signaling

**Subject Categories** Cancer; Computational Biology; Molecular Biology of Disease

**Mol Syst Biol. (2021) 17: e9730**

## Introduction

"Omics" technologies measure at the same time thousands of molecules in biological samples, from DNA, RNA, and proteins to metabolites. Omics datasets are an essential component of systems biology and are made possible by the popularization of analytical methods such as next-generation sequencing or mass spectrometry. Omics data have enabled the unbiased characterization of the molecular features of multiple human diseases, particularly in cancer (preprint: Jelinek & Wu, 2012; Iorio *et al*, 2016; Subramanian *et al*, 2017). It is becoming increasingly common to characterize multiple omics layers in parallel, with so-called "trans-omics analysis", to gain biological insights spanning multiple types of cellular processes (Sciacovelli *et al*, 2016; Kawata *et al*, 2018; Vitrinel *et al*, 2019). Consequently, many tools are developed to analyze such data (Tenenhaus *et al*, 2014; Argelaguet *et al*, 2018; Sharifi-Noghabi *et al*, 2019; Singh *et al*, 2019; Liu *et al*, 2019c), mainly by adapting and combining existing "single omics" methodologies to multiple

---

1  Faculty of Medicine, and Heidelberg University Hospital, Institute for Computational Biomedicine, Heidelberg University, Heidelberg, Germany
2  Faculty of Medicine, Joint Research Centre for Computational Biomedicine (JRC-COMBINE), RWTH Aachen University, Aachen, Germany
3  Faculty of Medicine, Institute of Experimental Medicine and Systems Biology, RWTH Aachen University, Aachen, Germany
4  Division of Nephrology and Clinical Immunology, Faculty of Medicine, RWTH Aachen University, Aachen, Germany
5  Department of Internal Medicine, Nephrology and Transplantation, Erasmus Medical Center, Rotterdam, The Netherlands
6  MRC Cancer Unit, Hutchison/MRC Research Centre, University of Cambridge, Cambridge, UK
7  Faculty of Health and Medical Sciences, Proteomics Program, Novo Nordisk Foundation Center for Protein Research, University of Copenhagen, Copenhagen, Denmark
8  Centre of Biological Engineering, University of Minho - Campus de Gualtar, Braga, Portugal
9  Department of Urology and Pediatric Urology, St. Antonius Hospital Eschweiler, Academic Teaching Hospital of RWTH Aachen, Eschweiler, Germany
10 Department of Urology and Kidney Transplantation, Martin Luther University, Halle (Saale), Germany
11 Department of Hematology, Erasmus MC, Rotterdam, The Netherlands
12 Institute for Research and Innovation in Health (i3s), Porto, Portugal
13 European Molecular Biology Laboratory, European Bioinformatics Institute (EMBL-EBI), Hinxton, UK
14 Molecular Medicine Partnership Unit, European Molecular Biology Laboratory, Heidelberg University, Heidelberg, Germany
   *Corresponding author. Tel: +44 1223 330608; E-mail: cf366@mrc-cu.cam.ac.uk
   **Corresponding author. Tel: +49 241 80 37750; E-mail: rkramann@ukaachen.de
   ***Corresponding author. Tel: +49 622 15451210; E-mail: julio.saez@uni-heidelberg.de
   †These authors contributed equally to this work
   ‡Present address: Cold Spring Harbor Laboratory,Cold Spring Harbor, NY, USA

---

parallel datasets. These methods identify groups of measurements and derive integrated statistics to describe them, effectively reducing the dimensionality of the datasets. These methods are useful to provide a global view of the data, but additional processing is required to extract mechanistic insights from them.

To extract mechanistic insights from datasets, some methods (such as pathway enrichment analysis) use prior knowledge about the players of the process being investigated. For instance, differential changes in the expression of the genes that constitute a pathway can be used to infer the activity of that pathway. Methods that a priori define groups of measurements based on known regulated targets (that we call footprints (Dugourd & Saez-Rodriguez, 2019)) of transcription factors (TFs; Alvarez *et al*, 2016; Garcia-Alonso *et al*, 2019), kinases/phosphatases (Wiredja *et al*, 2017), and pathway perturbations (Schubert *et al*, 2018) provide integrated statistics that can be interpreted as a proxy of the activity of a molecule or process. These methods seem to estimate more accurately the status of processes than classic pathway methods (Cantini *et al*, 2018; Schubert *et al*, 2018; Dugourd & Saez-Rodriguez, 2019). Since each of these types of footprint methods works with a certain type of omics data, finding links between them could help to interpret them collectively in a mechanistic manner. For example, one can use a network diffusion algorithm, such as TieDIE (Paull *et al*, 2013), to connect different omics footprints together (Drake *et al*, 2016). This approach provides valuable insights, but diffusion (or random walk) based algorithms do not typically take into account causal information (such as activation/inhibition) that is available and are essential to extract mechanistic information. TieDIE partially addressed this problem by focusing the diffusion process on causally coherent subparts of a network of interest, but it is thus limited to local causality.

Recently, we proposed the CARNIVAL tool (Liu *et al*, 2019b) to systematically generate mechanistic hypotheses connecting TFs through global causal reasoning supported by Integer Linear Programming. CARNIVAL connects activity perturbed nodes such as drug targets with deregulated TFs activities by contextualizing a signed and directed Prior Knowledge Network (PKN). We had hypothesized how such a method could potentially be used to connect footprint-based activity estimates across multiple omics layers (Dugourd & Saez-Rodriguez, 2019).

In this study, we introduce COSMOS (Causal Oriented Search of Multi-Omics Space). This approach connects TF and kinase/phosphatases activities (estimated with footprint-based methods) as well as metabolite abundances with a novel PKN spanning across multiple omics layers (Fig 1). COSMOS uses CARNIVAL's Integer Linear Programming (ILP) optimization strategy to find the smallest coherent subnetwork causally connecting as many deregulated TFs, kinases/phosphatases, and metabolites as possible. The subnetwork is extracted from a novel integrated PKN spanning signaling, transcriptional regulation, and metabolism of > 117,000 edges. CARNIVAL's ILP formulation effectively allows to evaluate the global network's causal coherence given a set of known TF, kinases/phosphatases activities and metabolite abundances. While we showcase this method using transcriptomics, phosphoproteomics, and metabolomics inputs, COSMOS can theoretically be used with any other additional inputs, as long as they can be linked to functional insights (for example, a set of deleterious mutations). As a case study, we generated transcriptomics, phosphoproteomics, and

metabolomics datasets from kidney tumor tissue and corresponding healthy tissue out of nine clear cell renal cell carcinoma (ccRCC) patients. We estimated changes of activities of TFs and kinase/phosphatases as well as metabolite abundance differences between tumor and healthy tissue. We integrated multiple curated resources of interactions between proteins, transcripts, and metabolites together to build a meta PKN. Next, we contextualized the meta PKN to a specific experiment. To do so, we identified causal pathways from our prior knowledge that connect the observed changes in activities of TFs, kinases, phosphatases, and metabolite abundances between tumor and healthy tissue. These causal pathways can be used as hypothesis generation tools to better understand the molecular phenotype of kidney cancer. We also refactored all functions to run the COSMOS approach into an R package.

## Results

### Building the multi-omics dataset

To build a multi-omics dataset of renal cancer, we performed transcriptomics, phosphoproteomics, and metabolomics analyses of renal nephrectomies and adjacent normal tissues of 11 renal cancer patients (for details on the patients see Dataset EV1). First, we processed the different omics datasets to prepare for the analysis. For the transcriptomics dataset, 15,919 transcripts with average counts > 50 were kept for subsequent analysis. In the phosphoproteomics dataset, 14,243 phosphosites detected in at least four samples were kept. In the metabolomics dataset, 107 metabolites detected across 16 samples were kept. Principal component analysis (PCA) of each omics dataset independently showed a clear separation of healthy and tumor tissues on the first component (transcriptomics: 40% of explained variance (EV), phosphoproteomics: 26% of EV, metabolomics: 28% of EV, Fig EV1), suggesting that tumor sample displayed molecular deregulations spanning across signaling, transcription and metabolism. Each omics dataset was independently submitted to differential (tumor vs. healthy tissue) analysis using LIMMA (Ritchie *et al*, 2015). Consistently with the PCA, a volcano plot overlapping the results of the differential analysis of each omics showed that the transcriptomics dataset led to larger differences and smaller *P*-values than phosphoproteomics and metabolomics extracted from the same samples (Fig EV2). This is further apparent by the number of hits under a given false discovery rate (FDR, Benjamini & Hochberg, 1995) threshold. We obtained 6,699 transcripts and 21 metabolites significantly regulated with FDR < 0.05. While only 11 phosphosites were found under 0.05 FDR, 447 phosphosites had an FDR < 0.2. This result confirmed the deep molecular deregulations of tumors spanning across signaling, transcription, and metabolism. Then, the differential statistics for all tested (not just the ones under the FDR threshold) transcripts, phosphopeptides, and metabolites were used for further downstream analysis, as explained below.

### Footprint-based transcription factor, kinase, and phosphatase activity estimation

We then performed computational footprint analysis to estimate the activity of proteins responsible for changes observed in specific

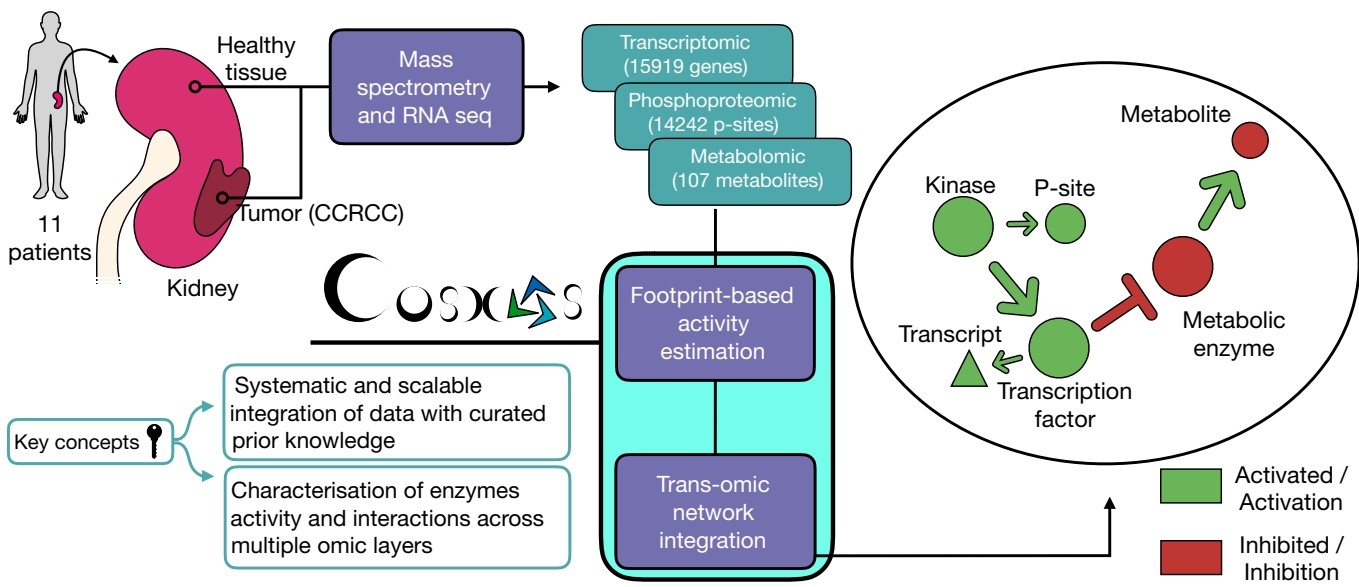

**Figure 1. Overview of analysis pipeline.**

From left to right: We sampled and processed 11 patient tumors and healthy kidney tissues from the same kidney through RNA-sequencing and 9 of those same patients through mass spectrometry to characterize their transcriptomics, phosphoproteomics, and metabolomics profiles. We calculated differential abundance for each detected gene, phosphopeptide, and metabolite. We estimated kinase and transcription factor activities using the differential analysis statistics and footprint-based methods. We used the estimated activities alongside the differential metabolite abundances to contextualize (i.e., extract the subnetwork that better explains the phenotype of interest) a generic trans-omics causal prior knowledge network (meta PKN).

omics datasets. By the term "activity", we refer to a quantifiable proxy of the function of a protein, estimated based on the footprint left by said activity. This definition can apply, but is not limited to, an enzyme's catalytic activity. Footprint-based activity estimation (Dugourd & Saez-Rodriguez, 2019) relies on the concept that the measured abundances of molecules (such as phosphopeptides or transcripts) can be used as a proxy of upstream (direct or indirect) regulator activities responsible for those changes (Rhodes *et al*, 2005; Casado *et al*, 2013; Ochoa *et al*, 2016). In the case of TF activity estimation, this means that measured changes in the abundances of transcripts give us information about the changes of activities of the transcription factors that regulate their abundance. An activity estimation only depends on the changes of the abundances measured in its target transcripts, not its own transcript abundance. In this study, we used the VIPER algorithm (Alvarez *et al*, 2016) to estimate the activity of transcription factors and kinases based on transcript and phosphopeptide abundances changes, respectively. For transcriptomics and phosphoproteomics data, this analysis estimates transcription factor and kinases/phosphatase activity, respectively. 24,347 transcription factors (TFs) to target interactions (i.e., transcript under the direct regulation of a transcription factor) were obtained from DoRothEA (Garcia-Alonso *et al*, 2019), a meta-resource of TF-target interactions. Those TF-target interactions span over 365 unique transcription factors. In parallel, 33,616 interactions of kinase/phosphate and their phosphosite targets (i.e., phosphopeptides directly (de)phosphorylated by specific kinases (phosphatases)) were obtained from OmniPath (Türei *et al*, 2016) kinase substrate network, a meta-resource focused on curated information on signaling processes. Only TFs and kinases/phosphatases with at least 10 and 5 detected substrates, respectively, were

included. This led to the activity estimation of 328 TFs and 174 kinases. In line with the results of the differential analysis, where fewer phosphosites were deregulated than transcripts, TF activities displayed a stronger deregulation than kinases. TF activity scores reached a maximum of 8.7 standard deviations (sd) for Transcription Factor Spi-1 Proto-Oncogene (*SPI1*) (compared to the null score distribution; sd compared to null is also referred to as a normalized enrichment score, NES), while kinase activity scores reached a maximum of 4.6 NES for Casein Kinase 2 Alpha 1 (*CSNK2A1*). In total, 102 TFs and kinases/phosphatase had an absolute score over 1.7 NES ($P < 0.05$) and were considered significantly deregulated in kidney tumor samples (Fig 2A). The presence of several known signatures of ccRCC corroborated the validity of our analysis. For instance, hypoxia (*HIF1A*), inflammation (*STAT2*, Fig 2B), and oncogenic (*MYC*, Cyclin Dependent Kinase 2 and 7 (*CDK2/7*, (Fig 2C)) markers were up-regulated in tumors compared to healthy tissues (Zeng *et al*, 2014; Schödel *et al*, 2016; Clark *et al*, 2020). Furthermore, among suppressed TFs we identified, the *HNF4A* gene has been previously associated with ccRCC (Lucas *et al*, 2005).

## Causal network analysis

We set out to find potential causal mechanistic pathways that could explain the changes we observed in TF, kinases/phosphatase activities, and metabolic abundances. Thus, we developed a systematic approach to search in public databases, such as OmniPath, for plausible causal links between significantly deregulated TFs, kinases/phosphatases and metabolites. In brief, we investigated if changes in TF, kinase/phosphatase activities, and metabolite abundance can explain each other with the support of literature-curated molecular

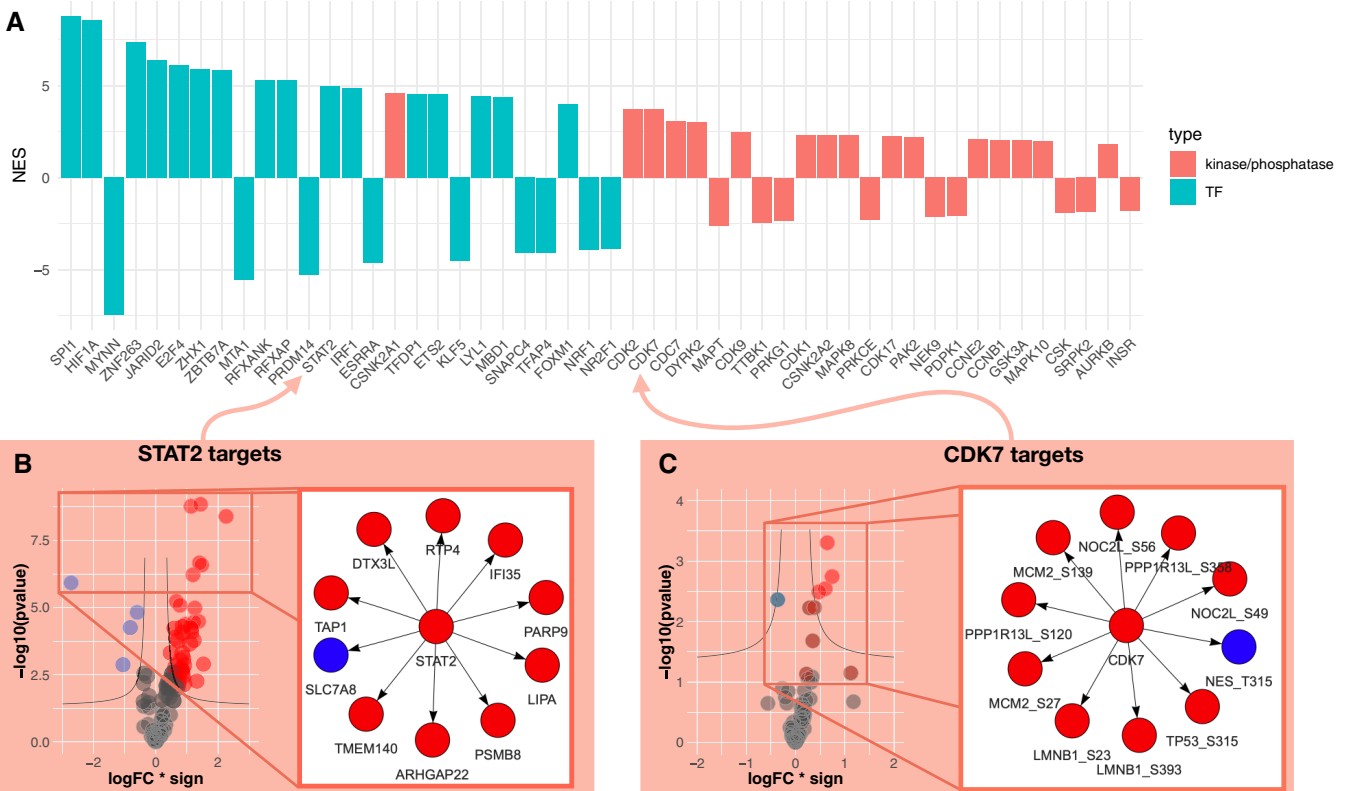

**Figure 2. Differentially regulated transcription factor, kinase, and phosphatase activities cancer vs. healthy tissue.**

A  Bar plot displaying the normalized enrichment score (NES, proxy of activity change) of the 25 up- or down-regulated TF and top 25 up- or down-regulated kinase and phosphatases activities between kidney tumor and adjacent healthy tissue.

B  Right panel shows the 10 most changing RNA abundances of the STAT2 regulated transcripts . Left panel shows the change of abundances of all STAT2 regulated transcripts that were used to estimate its activity change. *X*-axis represents log fold change of regulated transcripts multiplied by the sign of regulation (−1 for inhibition and 1 for activation of transcription). *Y*-axis represents the significance of the log fold change (−log$_{10}$ of *P*-value, LIMMA moderated unpaired *t*-test *P*-values). The black line is defined by the following function when fold change is negative : $y = abs(hAss - 1 + x/(x + vAss))$; and $y = abs(hAss - 1 + x/(x - vAss))$ when fold change is positive. abs() is the absolute value, hAss is the horizontal asymptote (hAss = 1.3) and vAss is the vertical asymptote (vAss = 0.3).

C  Right panel shows the 10 most changing phosphopeptide abundances of the CDK7 regulated phosphopeptides. Left panel shows the change of abundances of all CDK7 regulated phosphopeptides that were used to estimate its activity change. *X*-axis represents log fold change of regulated transcripts multiplied by the sign of regulation (−1 for inhibition and 1 for activation of transcription). *Y*-axis represents the significance of the log fold change (−log$_{10}$ of *P*-value, LIMMA moderated unpaired *t*-test *P*-values). The black line is defined by the following function when fold change is negative : $y = abs(hAss - 1 + x/(x + vAss))$; and $y = abs(hAss - 1 + x/(x - vAss))$ when fold change is positive. Where abs() is the absolute value, hAss is the horizontal asymptote (hAss = 1.3) and vAss is the vertical asymptote (vAss = 0.3).

interactions. An example of such a mechanism can be the activation of the transcription of *MYC* gene by *NFKB1*. Since both *NFKB1* and *MYC* display increased activities in tumors, and there is evidence in the literature that *NFKB1* can regulate *MYC* transcription (FANTOM4 database), it may indicate that this mechanism is responsible for this observation.

First, we needed to map the deregulated TFs, kinases, and metabolites on a causal prior knowledge network spanning over signaling pathways, gene regulation, and metabolic networks. Hence, we combined multiple sources of experimentally curated causal links together to build a meta causal prior knowledge network. This meta PKN must include direct causal links between proteins (kinase to kinase, TF to kinase, TF to metabolic enzymes, etc...), between proteins and metabolites (reactants to metabolic enzymes and metabolic enzymes to products) and between metabolites and proteins (allosteric regulations). High confidence (≥ 900

combined score) allosteric regulations of the STITCH database (Szklarczyk *et al*, 2016) were used as the source of causal links between metabolites and enzymes (Fig 3A). The directed signed interactions of the OmniPath database were used as a source of causal links between proteins (Fig 3B). The human metabolic network Recon3D (Brunk *et al*, 2018) (without cofactors and hyper-promiscuous metabolites, see Material and Methods) was converted to a causal network and used as the source of causal links between metabolites and metabolic enzymes (Fig 3C). The resulting meta PKN consists of 117,065 interactions and contains causal paths linking TFs/kinases/ phosphatases with metabolites and vice versa in a machine readable format. This network is available in the COSMOS R package.

We then used the meta PKN to systematically search causal paths between the deregulated TFs, kinases/phosphatases, and metabolites using an ILP optimization approach (see Material and Methods, Meta PKN contextualization). Here, we used CARNIVAL with our

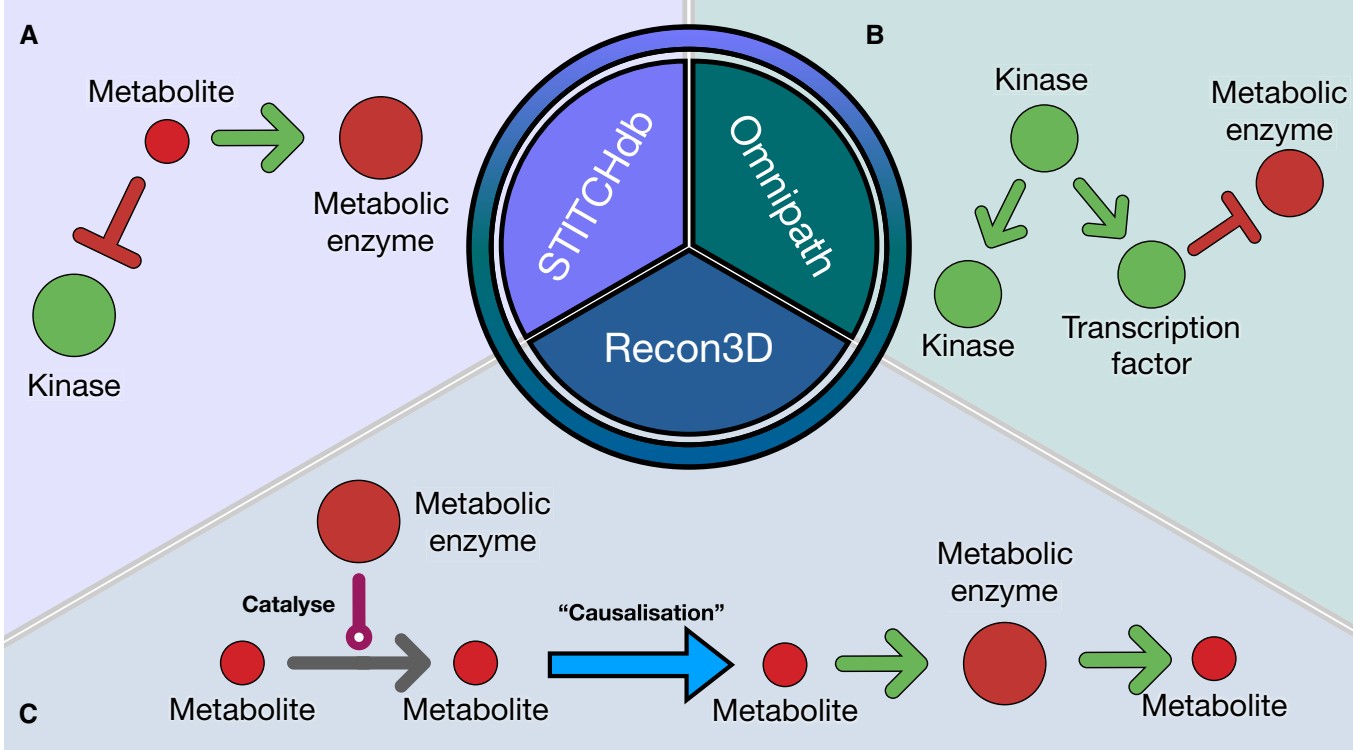

**Figure 3. Graphical explanation of meta PKN sources.**

A–C Schematic representation of the meta generic network (meta PKN) created combining STITCH, OmniPath and Recon3D. (A) STITCH provides information on inhibition/activation of enzyme activities mediated by metabolites. (B) OmniPath provides information inhibition/activation of enzyme activities mediated by other enzymes based mainly on curated resources. (C) Recon3D provides information on reactants and products associated with metabolic enzymes. To make this information compatible with the causal edges from OmniPath and STITCH, the interactions of recon3D are converted so that reactants "activate" their metabolic enzymes, which themselves "activate" their products.

meta PKN to find the smallest sign-coherent subnetwork connecting as many deregulated TFs, kinases/phosphatases, and metabolites as possible. First, we filtered out all interactions that do not involve genes expressed in our samples. Then, we removed nodes beyond a given number of steps downstream of inputs. We also removed any edge that leads to an incoherence between a TF activity score and the transcript abundance change of its targets (Appendix Fig S1A). We then performed a CARNIVAL run from TFs/kinases/phosphatases to metabolites to estimate the activity of TFs in the COSMOS solution network. These activities are used to filter out incoherent transcriptional regulation events from the meta PKN. Then, CARNIVAL is used to find causal paths going from TFs/kinases/phosphatases to the metabolites (the "forward network"). Finally, CARNIVAL is used to go from metabolites to TFs/Kinases/phosphatases ("backward network"). The choice of TFs/Kinases/phosphatases and metabolites to be included is detailed in Appendix Note 1. We combined the two networks (making union of the two sets edges and the union of the two sets of node attributes) to obtain a network with 449 unique edges (Appendix Fig S2, Dataset EV5). CARNIVAL finds a direct path connecting downstream measurements with upstream nodes, and thus, the solution networks do not contain loops. Loops can however appear in the final merged network when nodes are overlapping between "forward" and "backward" runs.

We then used our network to investigate the regulation of relevant signaling cascades and metabolic reactions in ccRCC. An overrepresentation analysis of the network solution nodes (with the hallmark genesets of MSigDB) displayed the interferon gamma (IFNg) response as the top significant pathway in our COSMOS network. Hence, we focused on the interaction members of this pathway (such as *NFKB1*, *HIF1A,* and *PNP*) and their crosstalks with metabolic deregulations to assess the relevance of the mechanistic hypotheses generated by COSMOS. We found that *NFKB1*, a central actor of the IFNg pathway is activated in ccRCC, consistently with other reports (Zhang *et al*, 2018; Rodrigues *et al*, 2018) where it was also demonstrated to be regulated by the *PI3K/AKT* pathway (An & Rettig, 2007). Interestingly, COSMOS also proposed the activation of *BCAT1*, one of the key enzymes of the branched-chain amino acid metabolism, orchestrated by *HIF1A* and *MYC* (Gordan *et al*, 2008; Ananieva & Wilkinson, 2018). Both mechanisms are shown in Fig 4 (1) and (2).

Of note, COSMOS provided deeper insights into these molecular mechanisms by linking *MYC* activation to *NFKB1*. The COSMOS model suggests that *MYC* up-regulates the expression of the metabolic enzyme *BCAT1*, potentially leading to the observed higher levels of glutamate, glutamine and reduced glutathione in ccRCC (marked as (2) in Fig 4). A strong role of *MYC* and glutamine metabolism in ccRCC development is known (Shroff *et al*, 2015).

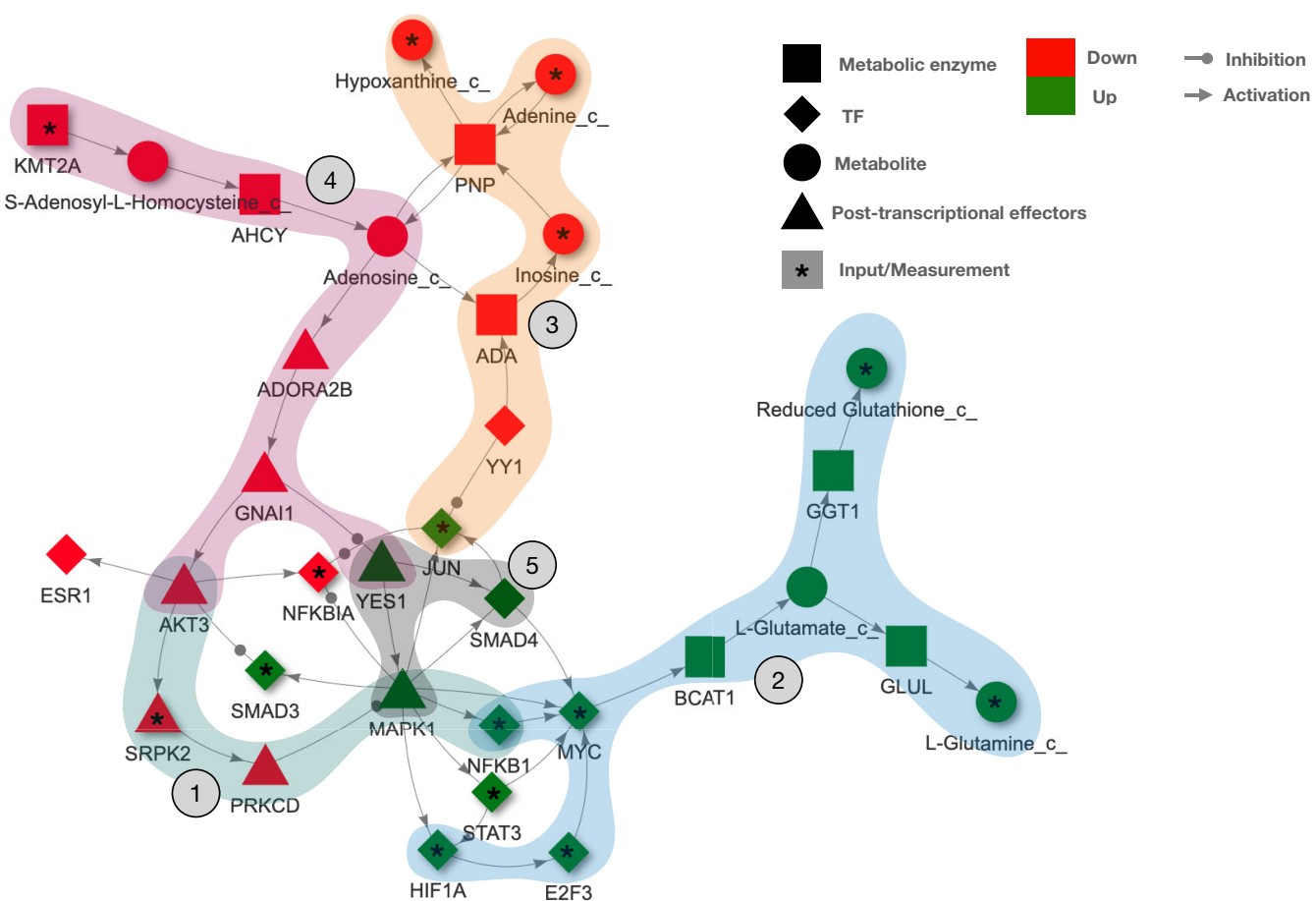

**Figure 4. COSMOS subnetwork centered on the interferon gamma response pathway.**

The figure includes the main members of the interferon gamma response pathway, the most enriched cancer hallmark in the full COSMOS network. We also display the metabolic enzymes that were hypothesized to be influenced downstream of this pathway, such as BCAT1 and PNP. The numbered mechanisms are discussed in the main text.

Consistently with what was hypothesized in a recent proteogenomics ccRCC study (Clark *et al*, 2020), we were able to capture crosstalks between members of the interferon gamma pathway (such as *JUN*), *YY1* and metabolic down-regulation observed in our data ((3) in Fig 4). COSMOS highlighted how *YY1* inhibition might be connected with the depletion of adenine, hypoxanthine, and inosine through regulation of the *ADA* and *PNP* metabolic enzyme (Popławski et al, 2017). The low levels of adenosine predicted by COSMOS might also be potentially linked to the down-regulation of *AKT3* and up-regulation of *YES1*, through a cascade which involves both *ADORA2B* and *GNAI1*, downstream of s-Adenosyl-L-homocysteine and inhibition of *KMT2A* ((4) in Fig 4). Finally, the COSMOS model showed a significant activation of *MAPK1* and *SMAD4* downstream of *YES1* (a member of the SRC family) ((5) in Fig 4).

## Consistency, robustness, and flexibility

Due to the combined effect of experimental noise and incompleteness of prior knowledge (kinase/substrate interactions, TF/targets interactions and meta PKN), it is critical to assess the performance of the pipeline presented above.

One way to estimate the performance is to check if the COSMOS mechanistic hypotheses correspond to correlations observed in tumor tissues (Appendix Fig S1B). Thus, on the one hand, a topology-driven co-regulation network was generated from the COSMOS network. The assumption behind this network is that direct downstream targets of the same enzymes should be co-regulated. On the other hand, a data-driven correlation network of TFs, kinases, and phosphatases was generated from tumor tissues alone. Assuming thresholds of absolute values of correlation ranging between 0 and 1 to define true positive co-regulations, the comparison between the topology-driven co-regulation network and the data-driven correlation network yielded a TPR ranging between 0.55 and 0 ($n = 269$ pairs of predicted/measured co-regulations) for the predictions (Appendix Fig S3). It performed consistently better than a random baseline (see Material and Methods) over the considered range of correlation coefficient thresholds. We also compared the results with network solutions obtained hiding either TFs or kinases/phosphatases. When TFs were hidden, COSMOS performed consistently better than the random baseline and reached a maximum TPR of 0.62. Of note, this curve was estimated from only $n = 21$ co-regulation events. When kinases and phosphatases were hidden, COSMOS

performed again consistently better than the random baseline and reached a maximum TPR of 0.58 ($n = 228$). In both cases, the performance of COSMOS was slightly larger than the full COSMOS performance (TPR = 0.55). This could be due to a lack of consistency across the omics data, although due to the low number of comparisons we could not make a conclusive statement. These results suggest that COSMOS' performance is relatively robust to removing either the phosphoproteomics or transcriptomics layers when trying to find connection between signaling and metabolism. However, using the three omics layers together yielded a larger network (367 edges (full) vs. 294 (hidden kinases) and 135 edges (hidden TFs)) and denser (1.67 edge/node ratio vs. 1.54 and 1.19 edge/node ratio, respectively) than when one omics layer was removed (Dataset EV3). Hence, using all layers yield a greater number of mechanistic hypotheses, even if not necessarily of higher quality.

To study the robustness of COSMOS to changes in the PKN, we generated a series of partially degraded PKN by randomly shuffling an increasing number of edges in the original PKN (2, 10, 20, 30, 40, 50% of all edges shuffled completely randomly). We ran COSMOS with each version of the PKN. We first compared the results of the "forward" COSMOS runs (connecting TFs and kinases with downstream metabolites). We calculated the absolute difference between the edge weight of the results (see Materials and Methods, meta PKN contextualization) obtained from each shuffled PKN with the result obtained from the original PKN. The edge weight represents the frequency of appearance (in %) of an edge across all the networks in the pool of network solutions. This showed that for the 2% shuffled network, the differences were relatively small (median of the absolute weight difference = 10), with 4% of edges flipped (i.e., 0 weight in shuffled network and 100 weight in original network, or vice versa). As expected, the differences were higher regarding the other shuffled networks, with medians weight differences of 10, 14, 23, 50, 28, and 35 for the 2, 10, 20, 30, 40, and 50% shuffled PKN, respectively (Appendix Fig S4A).

We then compared the results of the "backward" COSMOS runs (connecting metabolites with downstream TFs and kinases). Here, the comparison was far less quantitative because the optimization reported only a single solution for all runs except in the case of the 20% shuffled PKN (Appendix Fig S4B). 61, 31, 18, 12, 8, and 8% of edge weight differences were equal to 0 for the 2, 10, 20, 30, 40, and 50% shuffled PKN, respectively.

In both "forward" and "backward" runs, the network results had a relatively similar number of edges from the original and shuffled PKNs (min = 142, max = 342, mean = 263, SD = 63). The optimization thus consistently excluded a common set of edges covering the vast majority of the network, that contains over 56,000 edges.

We also compared the results we obtained from our samples with results obtained using another independent ccRCC dataset. We obtained the transcriptomics and phosphoproteomics dataset of the Clinical Proteomic Tumor Analysis Consortium (CPTAC) ccRCC patient cohort (Clark *et al*, 2020). Following the same approach as with our patient samples, we performed the differential analysis between tumor and healthy tissue for both omics datasets and estimated TFs and kinase/phosphatase activities. Then, we ran COSMOS to find mechanistic hypotheses explaining the connections between deregulated transcription factors and kinases/

phosphatases. The resulting COSMOS network was coherent with the results shown in the original publication and also provided additional information on the crosstalks between deregulated kinases and transcription factors. In particular, COSMOS captured the signaling crosstalks between *EGF*, *VEGF*, *AKT*, *MAPK*, *MTOR*, *NFKB*, and *MYC* (Dataset EV4). Finally, we compared which biological processes were captured in the COSMOS network generated from the data of our patient samples and the COSMOS network generated from the CPTAC ccRCC patient cohort. As shown in Fig EV3, the top over-represented pathways were very consistent between the two studies. Notably, *PI3K-AKT-MTOR* signaling and G2M checkpoint (Clark *et al*, 2020), *TNFA* signaling via *NFKB* (Al-Lamki *et al*, 2010), interferon gamma response (Thapa *et al*, 2013), *WNT* beta catenin signaling (Xu *et al*, 2016), and *IL6 JAK STAT3* signaling pathway were all significantly over-represented ($P < 0.02$).

Finally, we applied COSMOS to a public breast cancer dataset including transcriptomics and fluxomics measurements (Katzir *et al*, 2019) to connect signaling directly with metabolic flux estimation, instead of metabolite abundance measurements as done in the previous cases. We performed a differential analysis of transcript abundance and flux values between tumor cells cultured with and without glutamine. We then looked for mechanistic hypotheses connecting TF activity deregulations and changes in flux values. Coherently with the original study, almost all metabolites of the TCA cycle, glycolysis and pentose phosphate pathway were predicted to be down-regulated by COSMOS (Appendix Fig S5). Interestingly, COSMOS finds *HIF1A* as a master regulator of glycolysis through his effect on *HK1/2*, *GAPDH*, *GCK*, *ENO1*, and *LDHA* transcription. This is consistent with the known role of *HIF1A* in breast cancer (Samanta *et al*, 2014; Masoud & Li, 2015; Zhang *et al*, 2015; Singh *et al*, 2017). The down-regulation of *MYC* is also in line with the decreased activity of *HK2* and *LDHA* and *GLS1* enzymes which are important in aerobic glycolysis and glutamine catabolism (Dong *et al*, 2020).

## Discussion

In this paper, we present COSMOS, an analysis pipeline to systematically generate mechanistic hypotheses by integrating multi-omics datasets with a broad range of curated resources of interactions between protein, transcripts, and metabolites.

We first showed how TF, kinase, and phosphatase activities could be coherently estimated from transcriptomics and phosphoproteomics datasets using footprint-based analysis. This is a critical step before further mechanistic exploration. Indeed, transcript and phosphosite usually offer limited functional insights by themselves as their relationship with corresponding protein activity is usually not well characterized. Yet, they can provide information on the activity of the upstream proteins regulating their abundances. Thus, the functional state of kinases, phosphatases, and TFs is estimated from the observed abundance change of their known targets, i.e., their molecular footprint. Thanks to this approach, we could simultaneously characterize protein functional states in tumors at the level of signaling pathway and transcriptional regulation. Key actors of hypoxia response, inflammation pathway, and oncogenic genes were found to have especially strong alteration of their functional states, such as *HIF1A*, *EPAS1*, *STAT1/2*, *MYC,* and *CDK2*. Loss of

*VHL* is a hallmark of ccRCC and is directly linked to the stability of the *HIF* (*HIF1A* and *EPAS1*) proteins found deregulated by our analysis (Maxwell *et al*, 1999; Ivan *et al*, 2001; Jaakkola *et al*, 2001). Finding these established signatures of ccRCC to be deregulated in our analysis is a confirmation of the validity of this approach.

We then applied COSMOS with a novel meta causal Prior Knowledge Network spanning signaling, transcription, and metabolism to systematically find potential mechanisms linking deregulated protein activities and metabolite concentrations. To the best of our knowledge, this is the first attempt to integrate these three omics layers together in a systematic manner using causal reasoning. Previous methods studying signaling pathways with multi-omics quantitative datasets (Drake *et al*, 2016) connected TFs with kinases but they were limited by the preselected locally coherent subnetwork of the TieDIE algorithm. Introducing global causality along with metabolomics data allows us to obtain a direct mechanistic interpretation of links between proteins at different regulatory levels and metabolites. The goal of our approach is to find a coherent set of such mechanisms connecting as many of the observed deregulated protein activities and metabolite concentrations as possible. Using COSMOS is particularly interesting as all the proposed mechanisms between pairs of molecules (proteins and metabolites) have to be plausible not only in the context of their own pairwise interaction but also with respect to all other molecules that we wish to include in the model. For example, the proposed activation of *MYC* by *NFKB1* and *MAPK1* is further supported by *STAT3* activation, because *MAPK1* is also known to activate *STAT3*. Thus, we developed COSMOS to scale this type of reasoning up to the entire PKN with all significantly deregulated protein activities and metabolites. We relied on an ILP optimization through the CARNIVAL R package (Liu *et al*, 2019b) to contextualize this PKN with our data. We refined the optimization procedure to handle this very large PKN and built an R package to facilitate others to use it with their own data. Given a set of deregulated TFs, kinases/phosphatases, or metabolites, COSMOS provides the users with a set of coherent mechanistic hypotheses to explain changes observed in a given omics layer with upstream regulators from other omics layers. Thus, its aim is to integrate measured data with prior knowledge in a consistent and systematic manner, not to explicitly predict the outcome of new experiments.

Since the interferon gamma response pathway was the most over-represented cancer hallmark in the COSMOS network solution, we investigated further the relevance of the mechanistic hypothesis connecting members of this pathway. The network showed that the crosstalks between *MAPK1*, *NFKB1*, *MYC*, *HIF1A,* and *YY1* could explain the deregulation in glutamine and reduced glutathione metabolism, as well as inosine, hypoxanthine, and adenine. These were particularly relevant as they were important interactions in ccRCC. *MYC* and glutamine metabolism appear to be an interesting therapeutic target of ccRCC (Shroff *et al*, 2015). *YY1* is a known indirect inhibitor of *MYC* involved in cancer development (Austen *et al*, 1998). The COSMOS network showed *YY1* could also potentially have a role in the down-regulation of the *ADA* and *PNP* metabolic enzyme activities. Coherently, *PNP* has been shown to be non-essential in ccRCC cell lines, which is expected from down-regulated metabolic enzymes (Gatto *et al*, 2015). In addition, the link shown by COSMOS between *NFKB1* and *MYC* can have implications for the treatment of ccRCC, due to its pivotal role in arsenite (a drug used

in chemotherapy) treatment of cancer (Huang *et al*, 2014). Furthermore, the activation of the *NFKB1-MYC* link in *FBW7*-deficient cells seems to sensitize them to Sorafinib (a *MEK-Raf* inhibitor), a drug used in treatment of primary kidney cancer (Huang *et al*, 2014). In addition, *NFKB1* and *MYC* are both promising ccRCC treatment targets (Peri *et al*, 2013; Bailey *et al*, 2017). The link shown by COSMOS between *KMT2A* and adenosine is interesting, because *KMT2A* mutations have been reported in a number of ccRCC patients (Yan *et al*, 2019), suggesting that this enzyme might play a functional role in ccRCC development. Moreover, it has been proposed, at least in vitro, that ccRCC cell lines with low basal levels of phospho-*AKT* were sensitive to treatment with an adenosine analog (Kearney *et al*, 2015). The link between *YES1*, *MAPK1*, and *SMAD4* in the COSMOS network is especially relevant considering that *YES1* is a known targetable oncogene (Hamanaka *et al*, 2019). These examples illustrate the ability of COSMOS to extract mechanistic hypotheses to understand and potentially improve treatment of cancer by integration of multiple omics data and prior knowledge.

However, it is important to mention that COSMOS is only aimed at providing hypotheses to further explore experimentally. COSMOS does not aim at recapitulating all the molecular interactions that may be happening in a given context. Currently, COSMOS simply provides a large set of coherent mechanistic hypotheses, given the data and prior knowledge available. We argue that this facilitates the interpretation of a complex multi-omics dataset and guides the exploration of biological questions.

We assessed the performances and robustness of our approach. We computed a tumor specific correlation network of TF and kinase activities and compared it to the co-regulation predicted by COSMOS. This yielded encouraging results, though imperfect, underscoring the fact that the mechanisms proposed by COSMOS—like those by any similar tool—are hypotheses. It also highlighted that adding more omics data to integrate allows to generate more hypotheses and connect them together, but does not necessarily improve their predictive performances.

There are three main known limits to the predictions of COSMOS. First, the input data are incomplete. Only a limited fraction of all potential phosphosites and metabolites are detected by mass spectrometry. This means that we have no information on a significant part of the PKN; part of the unmeasured network is kept in the analyses and the values are estimated as intermediate "hidden values". Second, not all regulatory events between TFs, kinase, and phosphatases and their targets are known, and activity estimation is based only on the known regulatory relationships. Thus, many TFs, kinase, and phosphatases are not included because they have no curated regulatory interactions or no detected substrates in the data. Third, and conversely, COSMOS will find putative explanations within the existing prior knowledge that may not be the true mechanism.

These problems mainly originate from the importance that is given to prior knowledge in this method. Since prior knowledge is by essence incomplete, the next steps of improvement could consist of finding ways to extract more knowledge from the observed data to weight in the contribution of prior knowledge. For instance, one could use the correlations between transcripts, phosphosites and metabolites to quantify the interactions available in databases such as OmniPath. Importantly, any other omics that

relate to active molecules (such as miRNAs or metabolic enzyme fluxes) can be used to estimate protein activities through footprint approaches (such as DNA accessibility or PTMs other than phosphorylation) can be seamlessly integrated (as we showed with the fluxomic of the breast cancer dataset). Moreover, COSMOS was designed to work with bulk omics datasets, and it will be very exciting to find ways of applying this approach to single cell datasets. Encouragingly, the footprint methods that bring data into COSMOS seem fairly robust to the characteristics of single-cell RNA data such as dropouts (Holland *et al*, 2020). Related to the importance of prior knowledge, the PKN can also depend on how we interpret the information we have about molecular interactions.

In particular, we converted the reaction network of Recon3D into a causal network where metabolite reactants "activate" metabolic enzymes, and metabolic enzymes "activate" metabolite products. This first approximation assumes that metabolite abundances are only driven by their production rates. We plan to refine this in the future to include that metabolite abundances can change as a result of consumption as well. Finally, we expect that in the future, data generation technologies will increase coverage and our prior knowledge will become more complete, reducing the mentioned limitations. In the meantime, we believe that COSMOS is already a useful tool to extract causal mechanistic insights from multi-omics studies.

# Materials and Methods

### Reagents and Tools table

| Reagent/Resource | Reference or Source | Identifier or Catalog Number |
|---|---|---|
| **Chemicals, enzymes, and other reagents** | | |
| Guanidine hydrochloride | Sigma-Aldrich | G3272 |
| Ammonium carbonate | Merck | 1.59504.0250 |
| Ammonium hydroxide | Acros Organics | 255210010 |
| Optima™ LC/MS grade water | Fisher Scientific | W6-4 |
| Optima™ LC/MS grade acetonitrile | Fisher Scientific | A955-4 |
| SeQuant ZIC-pHILIC analytical column (150 × 2.1 mm, 5 μm) | Merck | 1.50460.0001 |
| SeQuant ZIC-pHILIC guard (20 × 2.1 mm, 5 μm) | Merck | 1.50438.0001 |
| Complete mini EDTA-free protease inhibitor cocktail | Roche | 04693124001 |
| 1.4 mm zirconium oxide beads | Bertin Technologies | KT03961-1-103.BK |
| 2.8 mm zirconium oxide beads | Bertin Technologies | KT03961-1-102.BK |
| Tris(2-carboxyethyl)phosphine | Sigma-Aldrich | C4706 |
| Chloroacetamide | Sigma-Aldrich | 22790 |
| Lys-C | Wako Chemicals | 129-02541 |
| Trizma base | Sigma-Aldrich | T1503 |
| Trypsin | Sigma-Aldrich | T6567 |
| C18 Sep-Pak Cartridges | Waters | WAT054955 |
| TMT10plex isobaric label reagent set | Thermo Fischer Scientific | 90406 |
| TMT11-131C Label Reagent | Thermo Fischer Scientific | A34807 |
| 5 μm Titansphere | GL Sciences | GS 502075000 |
| 2,5-dihydroxybenzoic acid | Sigma-Aldrich | 85707 |
| Empore C8 SPE Disks, 47mm | Empore | 66882-U |
| Ammonia solution 25% | Merck | 1054321011 |
| Ammonium bicarbonate | Sigma-Aldrich | 09830 |
| Acetonitrile | Merck | 1.00030.2500 |
| Formic Acid | Merck | 1.00264.1000 |
| Trifluoro acetic acid | Sigma-Aldrich | 8.08260.0501 |
| 1.9 μm Reprosil-Pur 120 C18 beads | Dr. Maisch | R119.aq.0003 |
| 75 μm ID capillary material | CM Scientific | TSP075375 |
| Guanidine hydrochloride | Sigma-Aldrich | G3272 |
| Complete mini EDTA-free protease inhibitor cocktail | Roche | 04693124001 |

**Reagents and Tools table**  (continued)

| Reagent/Resource | Reference or Source | Identifier or Catalog Number |
|---|---|---|
| 1.4 mm zirconium oxide beads | Bertin Technologies | KT03961-1-103.BK |
| 2.8 mm zirconium oxide beads | Bertin Technologies | KT03961-1-102.BK |
| Tris(2-carboxyethyl)phosphine | Sigma-Aldrich | C4706 |
| Chloroacetamide | Sigma-Aldrich | 22790 |
| Lys-C | Wako Chemicals | 129-02541 |
| Trizma base | Sigma-Aldrich | T1503 |
| Trypsin | Sigma-Aldrich | T6567 |
| C18 Sep-Pak Cartridges | Waters | WAT054955 |
| TMT10plex isobaric label reagent set | Thermo Fischer Scientific | 90406 |
| TMT11-131C Label Reagent | Thermo Fischer Scientific | A34807 |
| 5 μm Titansphere | GL Sciences | GS 502075000 |
| 2,5-dihydroxybenzoic acid | Sigma-Aldrich | 85707 |
| Empore C8 SPE Disks, 47mm | Empore | 66882-U |
| Ammonia solution 25% | Merck | 1054321011 |
| Ammonium bicarbonate | Sigma-Aldrich | 09830 |
| Acetonitrile | Merck | 1.00030.2500 |
| Formic Acid | Merck | 1.00264.1000 |
| Trifluoro acetic acid | Sigma-Aldrich | 8.08260.0501 |
| 1.9 μm Reprosil-Pur 120 C18 beads | Dr. Maisch | R119.aq.0003 |
| 75 μm ID capillary material | CM Scientific | TSP075375 |
| RNAeasyMini Kit | Qiagen | 74106 |
| KAPA RNA Hyperprep Kit | Roche | KR1351 |
| **Software** | | |
| MaxQuant 1.6.0.17 | https://maxquant.net/ | |
| Xcalibur QuanBrowser and QualBrowser (version 3.1) | Thermo Fisher Scientific | |
| R (version 4.0.2) | R Core Team (2020) | |
| CPLEX Optimizer (version 12.7.1.0) | IBM corp | |
| **Other** | | |
| Denator Stabilizor T1 | Denator | |
| Precellys 24 lysis & homogenization | Bertin Technologies | |
| Sonics Vibra-Cell Sonicator | Sonics & Materials | |
| SpeedVac Concentrator Plus | Eppendorf | |
| Nanodrop 2000 Spectrophotometer | Thermo Fischer Scientific | |
| Reversed-phase Acquity CSH C18 1.7 μm 1 × 150 mm column | Waters | |
| UltiMate 3000 high-pressure liquid chromatography (HPLC) system | Dionex | |
| Q Exactive HF-X | Thermo Fischer Scientific | |
| EASY-nLC 1200 UHPLC system | Thermo Fischer Scientific | |
| Denator Stabilizor T1 | Denator | |
| NovaSeq | Illumina | |
| Precellys 24 lysis & homogenization | Bertin Technologies | |
| Sonics Vibra-Cell Sonicator | Sonics & Materials | |
| SpeedVac Concentrator Plus | Eppendorf | |
| Nanodrop 2000 Spectrophotometer | Thermo Fischer Scientific | |
| Reversed-phase Acquity CSH C18 1.7 μm 1 × 150 mm column | Waters | |
| UltiMate 3000 high-pressure liquid chromatography (HPLC) system | Dionex | |

**Reagents and Tools table** (continued)

| Reagent/Resource | Reference or Source | Identifier or Catalog Number |
|---|---|---|
| Q Exactive HF-X | Thermo Fischer Scientific | |
| EASY-nLC 1200 UHPLC system | Thermo Fischer Scientific | |

## Methods and Protocols

### Sample collection and processing

We included a total of 22 samples from 11 renal cancer patients (6 men, age $65.0 \pm 14.31$, 5 women, age $65.2 \pm 9.257$ (mean $\pm$ SD)) for transcriptomics. Phosphoproteomics was also measured in a subset of 18 samples from 9 of these patients (6 men, age $65 \pm 14.31$; 3 women, age $63.33 \pm 11.06$ (mean $\pm$ SD)), and metabolomics was also measured in 16 samples from 8 out of these 9 patient (5 men, age $62 \pm 13.23$; 3 women, age $63.33 \pm 9.89$ (mean $\pm$ SD), Fig EV4, Dataset EV1). Patients underwent nephrectomy due to renal cancer. We processed tissue from within the cancer and a distant unaffected area of the same kidney. The tissue was snap-frozen immediately after nephrectomy within the operation room. The clinical data of the included patients is outlined in Dataset EV1. Histological evaluation showed clear renal cell carcinoma in all patients.

### Ethics

The local ethics committee of the University Hospital RWTH Aachen approved all human tissue protocols for this study (EK-016/17). The study was performed according to the declaration of Helsinki. Kidney tissues were collected from the Urology Department of the University Hospital Eschweiler from patients undergoing partial/- or nephrectomy due to kidney cancer. All patients gave informed consent.

### Human tissue processing

Kidney tissues were sampled by the surgeon from normal and tumor regions. The tissue was snap-frozen on dry-ice or placed in prechilled University of Wisconsin solution (#BTLBUW, Bridge to Life Ltd., Columbia, U.S.) and transported to our laboratory on ice.

### RNA Isolation, library preparation, NGS sequencing

RNA was extracted according to the manufacturer´s instructions using the RNeasy Mini Kit (QIAGEN). For rRNA-depleted RNA-seq using 1 and 10 ng of diluted total RNA, sequencing libraries were prepared with KAPA RNA HyperPrep Kit with RiboErase (Kapa Biosystems) according to the manufacturer's protocol. Sequencing libraries were quantified using quantitative PCR (New England Biolabs, Ipswich, USA). Equimolar pooling of the libraries was normalized to 1,4 nM, denatured using 0.2 N NaOH and neutralized with 400 nM Tris pH 8.0 prior to sequencing. Final sequencing was performed on a Novaseq6000 platform (Illumina) according to the manufacturer's protocols (Illumina, CA, USA).

### Metabolomics

Snap-frozen tissue specimens were cut and weighed into Precellys tubes prefilled with ceramic beads (Stretton Scientific Ltd., Derbyshire, UK). An exact volume of extraction solution (30% acetonitrile, 50% methanol, and 20% water) was added to obtain 40 mg specimen per mL of extraction solution. Tissue samples were lysed using a Precellys 24 homogenizer (Stretton Scientific Ltd., Derbyshire, UK). The suspension was mixed and incubated for 15 min at 4°C in a Thermomixer (Eppendorf, Germany), followed by centrifugation (16,000 *g*, 15 min at 4°C). The supernatant was collected and transferred into autosampler glass vials, which were stored at −80°C until further analysis.

Samples were randomized to avoid bias due to machine drift and processed blindly. LC-MS analysis was performed using a Q Exactive mass spectrometer coupled to a Dionex U3000 UHPLC system (both Thermo Fisher Scientific). The liquid chromatography system was fitted with a Sequant ZIC-pHILIC column (150 mm × 2.1 mm) and guard column (20 mm × 2.1 mm) from Merck Millipore (Germany) and temperature maintained at 45 °C. The mobile phase was composed of 20 mM ammonium carbonate and 0.1% ammonium hydroxide in water (solvent A), and acetonitrile (solvent B). The flow rate was set at 200 µL/min with the gradient described previously (Mackay *et al*, 2015). The mass spectrometer was operated in full MS and polarity switching mode. The acquired spectra were analyzed using XCalibur Qual Browser and XCalibur Quan Browser software (Thermo Scientific).

### Phosphoproteomics

Snap-frozen tissues were heat inactivated (Denator T1 Heat Stabilizor, Denator) and transferred to a GndCl solution (6 M GndCl, 25 mM Tris, pH 8.5, Roche Complete Protease Inhibitor tablets (Roche)) and homogenized by ceramic beads using 2 steps of 20 s at 5,500 rpm (Precellys 24, Bertin Technologies). The tissues were heated for 10 min at 95°C followed by micro tip sonication on ice and clarified by centrifugation (20 min, 16,000 *g*, 4°C). Samples were reduced and alkylated by adding 5 mM tris(2-carboxyethyl) phosphine and 10 mM chloroacetamide for 20 min at room temperature.

Lysates were digested by Lys-C (Wako) in an enzyme/protein ratio of 1:100 (w/w) for 1 h, followed by a dilution with 25 mM tris buffer (pH 8.5), to 2 M guanidine-HCl and further digested overnight with trypsin (Sigma-Aldrich; 1:100, w/w). Protease activity was quenched by acidification with TFA, and the resulting peptide mixture was concentrated on C18 Sep-Pak Cartridges (Waters). Peptides were eluted with 40% ACN followed by 60% ACN. The combined eluate was reduced by SpeedVac, and the final peptide concentration was estimated by measuring absorbance at *A*280 on a NanoDrop (Thermo Fisher Scientific). Peptide (300 µg) from each sample was labeled with 1 of 11 different TMT reagents according to the manufacturer's protocol (Thermo Fisher Scientific) for a total of four TMT sets. Each set comprised 10 samples and a common internal reference (composed of equal amounts of digested material from all samples).

After labeling, the samples were mixed and phosphopeptides were further enriched using titanium dioxide beads (5 µm

Titansphere, GL Sciences, Japan). TiO2 beads were pre-incubated in 2,5-dihydroxybenzoic acid (20 mg/ml) in 80% ACN and 1% TFA (5 μl/mg of beads) for 20 min. Samples were brought to 80% ACN and 5% TFA. 1.5 mg beads (in 5 μl of DHB solution) were added to each sample, which was then incubated for 20 min while rotating. After incubation, the beads were pelleted and fresh TiO2 beads were added to the supernatant for a second enrichment step. Beads were washed with five different buffers: (i) 80% ACN and 6% TFA, (ii) 10% ACN and 6% TFA, (iii) 80% ACN and 1% TFA, (iv) 50% ACN and 1% TFA, (v) 10% ACN and 1% TFA. The final washing step was performed on a C8 stage tip, from which the phosphopeptides were with 20 μl 5% NH4OH followed by 20 μl 10% NH4OH with 25% ACN. Eluted peptides were fractionated using a reversed-phase Acquity CSH C18 1.7 μm 1 × 150 mm column (Waters, Milford, MA) on an UltiMate 3000 high-pressure liquid chromatography (HPLC) system (Dionex, Sunnyvale, CA) operating at 30 μl/min. Buffer A (5 mM ammonium bicarbonate) and buffer B (100% ACN) were used. Peptides were separated by a linear gradient from 5% B to 35% B in 55 min, followed by a linear increase to 70% B in 8 min and 12 fractions were collected in a concatenated manner.

The peptide solution was adjusted in volume to an appropriate concentration and kept in loading buffer (5% ACN and 0.1% TFA) prior to autosampling. An in-house packed 15 cm, 75 μm ID capillary column with 1.9 μm Reprosil-Pur C18 beads (Dr. Maisch, Ammerbuch, Germany) was used with an EASY-nLC 1200 system (Thermo Fisher Scientific, San Jose, CA). The column temperature was maintained at 40°C using an integrated column oven (PRSO-V1, Sonation, Biberach, Germany) and interfaced online with a Q Exactive HF-X mass spectrometer. Formic acid (FA) 0.1% was used to buffer the pH in the two running buffers used. The gradients went from 8 to 24% acetonitrile (ACN) in 50 min, followed by 24–36% in 10 min. This was followed by a washout by a 1/2 min increase to 64% ACN, which was kept for 4.5 min. Flow rate was kept at 250 nL/min. Re-equilibration was done in parallel with sample pickup and prior to loading with a minimum requirement of 0.5 μl of 0.1% FA buffer at a pressure of 600 bar.

The mass spectrometer was running in data-dependent acquisition mode with the spray voltage set to 2 kV, funnel RF level at 40, and heated capillary at 275°C. Full MS resolutions were set to 60,000 at m/z 200 and full MS AGC target was 3E6 with an IT of 25 ms. Mass range was set to 375–1500. AGC target value for fragment spectra was set at 1E5, and intensity threshold was kept at 2E5. Isolation width was set at 0.8 m/z and a fixed first mass of 100 m/z was used. Normalized collision energy was set at 33%. Peptide match was set to off, and isotope exclusion was on.

Raw MS files were analyzed by MaxQuant software version 1.6.0.17 using the Andromeda search engine. Proteins were identified by searching the higher-energy collisional dissociation (HCD)–MS/MS peak lists against a target/decoy version of the human UniProt protein database (release April 2017) using default settings. Carbamidomethylation of cysteine was specified as fixed modification, and protein N-terminal acetylation, oxidation of methionine, pyro-glutamate formation from glutamine, and phosphorylation of serine, threonine, and tyrosine residues were considered as variable modifications. The "maximum peptide mass" was set to 7,500 Da, and the "modified peptide minimum score" and "modified maximum peptide score" were set to 25. Everything else was set to default values. The mass spectrometry proteomics data have been deposited to the ProteomeXchange Consortium via the PRIDE partner repository. Data are available via ProteomeXchange with identifier PXD018218 with the following reviewer account details: Username: reviewer81921@ebi.ac.uk Password: Kidney2020.

### Data normalization and differential analysis

In the phosphoproteomics dataset, 19285 unique phosphosites were detected across 18 samples. Visual inspection of the raw data PCA first 2 components indicated two major batches of samples (1st batch : "38KI", "38TU", "15KI", "15TU", "29KI", "29TU", "16KI", "16TU", "32KI", "32TU", "35KI", "35TU"; 2nd batch : "40KI", "40TU", "24KI", "24TU", "11KI", "11TU"). Thus, each batch was first normalized using the VSN R package (Huber *et al*, 2002; Välikangas *et al*, 2018). We removed p-sites that were detected in < 4 samples, leaving 14,243 unique p-site to analyze. Visual inspection of the PCA first two components of the normalized data revealed that the first batch of samples could itself be separated in 3 batches (4 batches across all samples). Thus, we used the removeBatchEffect function of LIMMA to remove the linear effect of the 4 batches. Differential analysis was performed using the standard sequence of lmFit, contrasts.fit and eBayes functions of LIMMA, with FDR correction.

For the transcriptomics data, counts were extracted from fast.q files using the RsubRead R package and GRCh37 (hg19) reference genome. Technical replicates were averaged, and genes with average counts under 50 across samples were excluded, leaving 15919 genes measured across 22 samples. To allow for logarithmic transformation, 0 count values were scaled up to 0.5 (similar to the voom function of LIMMA). Counts were then normalized using the VSN R package function and differential analysis was performed with LIMMA package, in the same way as the phosphoproteomics data.

For the metabolomics data, 107 metabolites were detected in 16 samples. Intensities were normalized using the VSN package. Differential analysis was done using LIMMA in the same manner as for phosphoproteomics and transcriptomics. All data are available at: https://github.com/saezlab/COSMOS.

### Footprint-based analysis

TF-target collection was obtained from DoRothEA A,B,C and D interaction confidence levels from the DoRothEA R package (version 1.1.0). For the enrichment analysis, the viper algorithm (Alvarez *et al*, 2016) was used with the LIMMA moderated t-value as gene level statistic (Zyla *et al*, 2017). The eset.filter parameter was set to FALSE. Only TFs with at least 25 measured transcripts were included.

Kinase substrate collection was obtained using the default resource collection of OmniPath, with the URL "http://omnipathdb.org/ptms?fields = sources,references&genesymbols = 1" (version of 2020 Feb 05). For the enrichment analysis, the viper algorithm was used with the LIMMA moderated *t*-value as phosphosite level statistic. The eset.filter parameter was set to FALSE. Only TFs with at least 5 measured transcripts were included. All data are available at https://github.com/saezlab/COSMOS_MSB/tree/main/data.

### Meta PKN construction

To propose mechanistic hypotheses spanning through signaling, transcription and metabolic reaction networks, multiple types of interactions have to be combined together in a single network.

Thus, we built a meta Prior Knowledge Network (PKN) from three online resources, to incorporate three main types of interactions. The three types of interactions are protein–protein interactions, metabolite-protein allosteric interactions, and metabolite-protein interactions in the context of a metabolic reaction network. Protein–protein interaction was imported from OmniPath with the URL http://omnipathdb.org/interactions?types = post_translational,transcriptional&datasets = omnipath,pathwayextra, dorothea&fields = sources,references,curation_effort,dorothea_level,type&genesymbols = yes (version of 2020 July 17), and only signed directed interactions were included (is_stimulation or is_inhibition columns equal to 1). Metabolic-protein allosteric interactions were imported from the STITCH database (version of 2019 November 06), with combined confidence score ≥ 900 after exclusion of interactions relying mainly on text mining.

For metabolic-protein interactions in the context of metabolic reaction network, Recon3D was downloaded from https://www.vmh.life/#downloadview (version of 2019 Feb 19). Then, the gene rules ("AND" and "OR") of the metabolic reaction network were used to associate reactants and products with the corresponding enzymes of each reaction. When multiple enzymes were associated with a reaction with an "AND" rule, they were combined together as a single entity representing an enzymatic complex. Then, reactants were connected to corresponding enzymatic complexes or enzymes by writing them as rows of simple interaction format (SIF) table of the following form: reactant;1;enzyme. In a similar manner, products were connected to corresponding enzymatic complexes or enzymes by writing them as rows of simple interaction format (SIF) table of the following form: enzyme;1;product. Thus, each row of the SIF table represents either an activation of the enzyme by the reactant (i.e., the necessity of the presence of the reactant for the enzyme to catalyze it's reaction) or an activation of the product by an enzyme (i.e., the product presence is dependent on the activity of its corresponding enzyme). Most metabolite–protein interactions in metabolic reaction networks are not exclusive, thus measures have to be taken in order to preserve the coherence of the reaction network when converted to the SIF format. First, metabolites that are identified as "Coenzymes" in the Medical Subject Heading Classification (as referenced in the PubChem online database) were excluded. Then, we looked at the number of connections of each metabolite and searched the minimum interaction number threshold that would avoid excluding main central carbon metabolites. Glutamic acid has 338 interactions in our Recon3D SIF network and is the most connected central carbon metabolite, thus any metabolites that had more than 338 interactions was excluded. An extensive list of Recon3D metabolites (PubChem CID) with their corresponding number of connections is available in Dataset EV2. Metabolic enzymes catalyzing multiple reactions were uniquely identified for each reaction to avoid cross-links between reactants and products of different reactions. Finally, exchange reactions were further uniquely identified according to the relevant exchanged metabolites, as to avoid confusion between transformation of metabolites and simply exchanging them between compartments.

Finally, each network (protein–protein, allosteric metabolite–protein, and reaction network metabolite–protein) was combined into a single SIF table. This network is available in the COSMOS R package.

## Meta PKN contextualization

COSMOS uses the CARNIVAL R package to perform the network optimization via an ILP algorithm. In brief, we try to minimize the value of an objective function that depends on two main factors: (i) the mismatch between the simulated values of kinases, TFs, and metabolites for a given causal network and the corresponding available values estimated from the measurements and (ii) the size of the solution network. For each run, given the prior knowledge network and the input and measurements, a set of constraints are generated to define the solution space (based on the objective function) that the ILP solver (IBM CPLEX in our study) explores to find an optimal solution (Melas *et al*, 2015; preprint: Liu *et al*, 2019a). After a given amount of time (decided by the user), the search is stopped and the best solution at this point is returned by CPLEX. The solution is typically a pool (or family) of networks that are all equally optimal with respect to the objective function. Thus, CARNIVAL reports the solution as a set of edges with an associated weight that represent their frequency of appearance in the current network pool. CARNIVAL needs a set of starting and end nodes to look for paths in between. TFs, kinases, and phosphatases absolute normalized enrichment scores greater than 1.7 standard deviation were considered deregulated. Coherently, metabolites with uncorrected *P*-values smaller than 0.05 were considered deregulated. We give more information on the rational to choose an appropriate threshold in the Appendix Note 1. This yielded a set of 98 TFs, 25 kinases/phosphatase, and 41 metabolites to be used as input and measurements for COSMOS.

Then, the PKN is pre-processed in three steps to make it easier for CARNIVAL to find a solution network, as detailed below.

### Filtering

The generic meta PKN contained 117,065 edges. We first filtered the meta PKN to keep only genes that are expressed. With the main dataset presented in this paper, we considered the 15,919 genes that remained after removing the lowly expressed genes (defined as those with average count under 50 across the 22 samples, based on the count distribution) as expressed. This reduced the size of the meta PKN from 117,065 edges to 66,749 edges.

### Reduction

At this stage, the meta PKN may contain independent network modules that do not include any of the actual input nodes (the significant TF, kinase/phosphatase activities, and metabolites). Thus, we filter out any gene that cannot be connected to any input node. We define a maximum given number of steps to avoid excessively long causal paths that would be un-plausible and thus have unclear biological relevance. We chose 8 steps downstream of signaling inputs for the "forward" run (signaling to metabolism) and 7 steps downstream of metabolic inputs for the "backward" run (metabolism to signaling) as > 90% of the PKN could be captured in that number of steps.

### Correction

We use the transcriptomics data differential gene expression analysis results to directly remove any edge that leads to an incoherence between a TF activity and its target transcript abundance change (which is a wrongly predicted transcriptional regulation event). This is done once before running CARNIVAL, using TF activities predicted with DoRothEA. Then, we do a pre-run of CARNIVAL (TFs/kinases/phosphatases -> metabolites) to generate a first

solution network. We can subsequently use TF activities predicted by CARNIVAL to filter out any wrongly predicted transcriptional regulation event from the meta PKN (Appendix Fig S3A).

Then, we first set the deregulated kinases, phosphatases, and TFs as starting points and deregulated metabolites as end points ("forward" run). This direction represents regulations first going through the signaling and transcriptional part of the cellular network and stops at deregulated metabolites in the metabolic reaction network. However, since metabolite concentration can also influence the activity of kinases and TFs through allosteric regulations, we also ran CARNIVAL by setting deregulated metabolites as starting points and deregulated TFs, kinases, and phosphatases as end points ("backward" run). The "forward" run was performed with a time limit of 7,200 s and yielded a network of 162 edges. The "backward" run was performed with a time limit of 21,800 s and yielded a network of 302 edges.

There was a single incoherence in the predicted sign of *ARNT2* transcription factor (−1 in "forward" run, 1 in "backward" run) between the common part of the two resulting networks. We made the union of the two networks, resulting in a combined network of 449 unique edges, while preserving the incoherent sign of *ARNT2* in the corresponding node attributes of the network (Dataset EV5).

### Coherence between COSMOS mechanistic hypotheses and omics measurements

To assess the robustness of COSMOS predictions, we compared co-regulations predicted by the COSMOS solution network with co-regulations estimated from correlation between kinase, phosphatase, and TF activities. When multiple nodes are co-regulated by a common parent node in the COSMOS network, we can assume that the activity of the co-regulated nodes should be correlated. Thus, we created a correlation network with the TF and kinase/phosphatase activities estimated at a single sample level. To estimate the single sample level activities, normalized RNA counts and phosphosite intensities were scaled (minus mean over standard deviation) across samples. Thus, the value of each gene and phosphosite is now a z-score relative to an empirical distribution generated from the measurements across all samples. We used these z-scores as input for the viper algorithm to estimate kinase/phosphatases and TF activities at single sample level. Thus, the resulting activity scores in a sample are relative to all the other samples. Then, a correlation network was built using only tumor samples. Thus, the correlation calculated this way represents co-regulations that are supported by the available data in tumor. We defined the ground truth for co-regulations as over a range of absolute correlation coefficients between 0 and 1 with a 0.01 step. Thus, a True Positive here is a co-regulation predicted from the topology of the COSMOS network that also has a corresponding absolute correlation coefficient in tumor samples above the given threshold. Since defining a ground truth in such a manner can yield many false positives (a correlation can often be spurious), the TPR of COSMOS was always compared to a random baseline. This approach was repeated for COSMOS solution networks obtained after hiding either kinase and phosphatases or TFs.

### Robustness analysis

We generated a series of subsets of the original meta PKN where increasing amounts of interactions are shuffled randomly. Starting from the full meta PKN, we shuffled 2, 10, 20, 30, 40, and 50% of interactions. Each shuffling is independent from the others (the missing interactions are all selected randomly at each percentage case). Then, COSMOS was run for each meta PKN subset with the same parameter as the original run.

### CPTAC ccRCC data analysis

The CPTAC ccRCC transcriptomics and phosphoproteomics datasets of the proteogenomics study of ccRCC (Clark *et al*, 2020) were obtained from the CPTAC data portal. We kept 20,284 phosphosites that were detected across at least 10% of the 185 patient and healthy samples (110 and 75, respectively).

We filtered out lowly expressed genes (RPKM (Reads Per Kilobase of transcript, per Million mapped reads) < 170, based on the inflexion point observed in the RPKM distribution) from the transcriptomics dataset, keeping 14,921 genes for further analysis.

LIMMA was used for both phosphoproteomics and transcriptomics to perform a differential analysis between healthy and tumor samples.

Kinase and transcription factor activities were performed with the same parameters as with our own ccRCC patient samples (see footprint-based analysis). 57 kinases and 97 TFs with absolute NES > 1.7 were used as input and measurements in the COSMOS pipeline. The meta PKN was reduced to keep only nodes with a maximum distance of 8 steps downstream of input kinases and TFs. The kinase to TF CARNIVAL run was performed with a time limit of 7,200 s. The TF to kinase run was performed with a time limit of 21,800 s. The union of the "forward" and "backward" run networks resulted in a final COSMOS network of 480 edges.

### Breast cancer data analysis

Multi-omics experimental data for breast cancer cell lines was obtained from (Katzir *et al*, 2019). The authors performed experimental measurements on the MCF7 cell line under normal growth conditions, glutamine deprivation, and oligomycin supplementation.

We obtained mRNA expression quantification of 1,905 metabolic genes and filtered those whose mean across all conditions was at least 0.1% of the maximum observed expression value. The experiments were split in 2 batches, leading us to regress this effect out. We then fit a linear model using the LIMMA package, from which we obtained t-statistic values at the gene level for a given comparison pair. Finally, TF activity scores were estimated using regulon confidence A, B, and C with a minimum of 25 targets per TF with the VIPER package, using the pleiotropy correction.

Fluxomics measurements estimated from $^{13}$C-assisted metabolomics were available for 44 metabolic reactions included in the Recon 3D genome-scale metabolic model. We computed the $\log_2$ fold change between each pair of conditions to be analyzed.

COSMOS was then used to generate context-specific subnetworks using the transcription factor NES and the fluxomics $\log_2$ fold change as inputs and measurements. It was run without using the correction and reduction step, with a time limit of 7,200 s on the "forward" and "backward" runs.

## Data availability

All code used in this study is available at: https://github.com/saezlab/COSMOS_MSB.

- The COSMOS R package is available at: https://github.com/saezlab/COSMOS.
- Processed data used in this study is available at: https://github.com/saezlab/COSMOS_MSB/tree/master/data.
- RNA-seq counts are available: https://github.com/saezlab/COSMOS_MSB/blob/main/data/RNA_transcriptomic_raw.csv.
- Phosphoproteomics raw data is available via ProteomeXchange with identifier PXD018218.
- Metabolomic data is available at: : https://github.com/saezlab/COSMOS_MSB/tree/main/data/original_metab_data.
- The meta PKN used in this study is available via the COSMOS R package (https://github.com/saezlab/COSMOS).

**Expanded View** for this article is available online.

## Acknowledgements

A.D. and E.G. were Marie-Curie Early Stage Researchers supported by the European Union's Horizon 2020 research and innovation program (675585 Marie-Curie ITN "SymBioSys") to J.S.R. A.D. was funded by German Federal Ministry of Education and Research (Bundesministerium für Bildung und Forschung BMBF) MSCoreSys research initiative research core SMART-CARE (031L0212A). This work was further supported by the JRC for Computational Biomedicine which was partially funded by Bayer AG, and the Medical Research Council (MC_UU_12022/6 to C.F. and M.S.). The Novo Nordisk Foundation Center for Protein Research is supported by Novo Nordisk Foundation grant number NNF14CC0001. J.V.O. was funded by a grant from Danish Council for Independent Research (8020-00100B) to partly support K.B.E. who was also supported in part by the Lundbeck Foundation (R193-2015-243). R.K. was supported by grants of the German Research Foundation (DFG: SFBTRR57, P30; SFBTRR219 C05, CRU344, P1), by a Grant of the European Research Council (ERC-StG 677448), a Grant of the State of North Rhine-Westphalia (Return to NRW), the BMBF eMed Consortia Fibromap, the ERA-CVD Consortia MEND-AGE, the Else Kroener Fresenius Foundation (EKFS) and the Interdisciplinary Centre for Clinical Research (IZKF) within the faculty of Medicine at the RWTH Aachen University (O3-11). C.K. was supported by the German Society of Internal Medicine (DGIM). Thanks to Hyojin Kim for her contribution to the original COSMOS logo design. Thanks to Denes Turei for his help with putting the meta PKN online. We thank E. Ruppin and R. Katzir for helping us with the breast cancer dataset from Katzir *et al* (2019). Open Access funding enabled and organized by ProjektDEAL.

## Authors contributions

AD and JS-R designed the method. AD coded the pipeline and ran the analysis. AD with input from CK, MS, CF and JS-R. interpreted results. CK collected the patient samples and generated RNA-Seq libraries. EG developed and adapted CARNIVAL to the pipeline. KBE, DBB-J and JVO generated the phosphoproteomics dataset. EMJB performed final RNA-sequencing on Novaseq6000 platform. MS and ASHC performed the liquid chromatography mass spectrometry-based metabolomics analyses and processed the data. JK collected patient consents and samples. CF, RK and JS-R supervised the project. AD wrote the manuscript with help from JS-R. AS and PB processed the CPTAC data. VV and MR analyzed the breast cancer dataset. AG has developed the R package version of the method.

## Conflict of interest

The authors declare that they have no conflict of interest. C.F. is a member of the scientific advisory board of Owlstone and scientific adviser of Istesso. JSR has received funding from GSK and expects to receive funding from Sanofi and consultant fees from Travere Therapeutics.

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
