## [Review Process File · Molecular Systems Biology]

Causal integration of multi-omics data with prior knowledge to generate mechanistic hypotheses

Aurelien Dugourd, Christoph Kuppe, Marco Sciacovelli, Enio Gjerga, Kristina Bennet Emdal, Dorte Breinholdt Bekker-Jensen, Jennifer Kranz, Eric Bindels, Sofia Costa, Jesper Olsen, Christian Frezza, Rafael Kramann, Julio Saez-Rodriguez, Attila Gábor, Vitor Vieira, Abel Sousa, Pedro Beltrao, and Miguel Rocha

DOI: 10.15252/msb.20209730

Corresponding author(s): Julio Saez-Rodriguez (julio.saez@bioquant.uni-heidelberg.de), Christian Frezza (CF366@MRC-CU.cam.ac.uk), Rafael Kramann (rkramann@ukaachen.de)

Review Timeline:	Submission Date:	19th May 20
	Editorial Decision:	1st Jul 20
	Revision Received:	9th Nov 20
	Editorial Decision:	14th Dec 20
	Revision Received:	18th Dec 20
	Accepted:	21st Dec 20

Editor: Maria Polychronidou

Transaction Report:

Thank you again for submitting your work to Molecular Systems Biology. We have now heard back from two of the three referees who agreed to evaluate your study. Unfortunately, after a series of reminders we did not manage to obtain a report from reviewer #2. In the interest of time, we have decided to proceed with making a decision based on the two available reports. Overall, the reviewers recognize that the presented framework seems interesting. However, they raise a series of concerns, which we would ask you to address in a major revision.

Without repeating all the points listed below, some of the more fundamental issues are the following.

REFEREE REPORTS

Reviewer #1:

In this manuscript, Dugourd et al. presents a method named COSMOS that integrates transcriptome, phosphoproteome and metabolome data into a directed network representing prior knowledge on transcription regulation, signaling transduction via phosphorylation, and cellular metabolism. The authors applied this method to a multi-omic dataset from a total of 11 renal cell carcinoma patients, and demonstrated that COSMOS was able to generate meaningful and reasonable biological hypotheses. The method seems to be a natural extended application of the CARNIVAL algorithm that the authors have developed previously to multi-omic data and multi-omic prior knowledge network (PKN). However, the authors has pointed out that this methodological framework can in theory be applied to any types of multi-omic data where the corresponding PKN can be reliably constructed, which makes COSMOS interesting for potentially broader applications. Overall we feel that the manuscript is clearly written and the results quite interesting. Our specific comments are as follows.

Major comments:

1. One key novelty of the COSMOS method over the previous CARNIVAL method is that COSMOS extends the application to multi-omic data. Therefore it can be important and valuable to show that the added domains of omic data can really lead to improved performance (e.g. accuracy of the predictions) over a single omic approach. The rationale of this comment is that due to technical limitations and noise in data, different omic data can have very poor correlation or consistency, and adding further omic data may not lead to a significant gain in practise. Specifically, for example, comparing the COSMOS prediction using the full data and PKN (involving three types of omic data) to that using only two of the three types of omic data, does the former have higher true-positive rate as measured from the consistency analysis the authors used?
2. One key novelty and potential value of COSMOS (and CARNIVAL) over common pathway enrichment or the various "footprint-based" methods is that COSMOS can infer "causal" links that represent hypotheses about the potential mechanisms of actions underlying the observed biology, which can be further investigated using experiments. We understand that additional experimental validation may be out of the scope of this study, but we think that it is highly desirable if the authors can try to provide more direct validations of their inferred causal "paths", beyond the consistency analysis the authors have shown. The may be acheived using existent data. For example, it can be possible to mine pulic datasets where a input node or "upper-stream" node gene is knocked out or inhibited with drugs, and demonstrate with such data that some of the downstream node changed in the direction as predicted by COSMOS.
3. The authors mentioned that the method relies on reliable sources of prior knowledge to construct a decent PKN. Therefore the important question is how robust the predictions are subject to the quality of the PKN? It may be desirable to perform a robustness analysis by, e.g. removing a certain fraction of the edges from the PKN and assess how much the results are affected.
4. In the metabolomic part of the PKN construction, when a metabolic enzyme X catalyzed the conversion of metabolite A to metabolite B, the current network structure representing this is "A 

X  B". However, another equally reasonable "causal" path is "B --| X --| A" (i.e. the product B inhibits the enzyme X in a negative feedback via allosteric regulation; and X consumes the reactant A, therefore effectively represented by an inhibition edge). Both paths may be added to the PKN without additional prior knowledge on which actually happens.

5. In the section, 'Building the multi-omics dataset', the authors concluded that TF dysregulation is more pervasive than signaling, transcription, and metabolism. Firstly, it is not explained well how the author came to the conclusion. Secondly, it may help strengthen this claim by a plot of fold change (or Effect Size) for each gene for all the transcripts, metabolites, and phosphites.

Minor comments:

1. Does the PKN contain cycles (i.e. loops) that are inconsistent within itself (i.e. contain inconsistent signs)? If yes, does such structures pose a problem in the computation or interpretation of results, and how are they handled in the method? If not, are these structures removed explicitly and how?

2. We feel that the Methods need to be expanded to provide additional necessary details. For example, how was the integer programming (ILP) part of the COSMOS algorithm formulized? It may help to explicitly describe the ILP formulation of the problem. The part on the "footprint-based" inference of transcription factor (TF)/kinase/enzyme activity also needs further clarification -- to readers not familiar with such methods, it is not clear how this works or what algorithm(s) is/are used for TF/kinase/enzyme.

3. In Figure 4, the "Edge arrow shape" and the corresponding "Effect" legend seems to be wrong?

Reviewer #3:

Summary

The authors developed COSMOS, a novel Prior Knowledge Network (PKN), that uses a previously published tool, CARNIVAL, to generate networks using transcriptomics, metabolomics, and phosphoproteomics data. They applied this tool to a dataset of tumor and normal tissue samples from 11 ccRCC patients. Their work identifies transcription factors, kinases, and metabolites that are dysregulated in ccRCC and, using COSMOS/CARNIVAL, generate a network that suggests novel mechanistic hypotheses. The authors have generated a useful PKN and tool for analyzing a novel combination of omics data. However, the utility of COSMOS/CARNIVAL as a tool for generally combining multiple omics datasets is unclear. This study may contribute novel understanding of ccRCC biology, however, further validation of their network is necessary to validate both the hypotheses their network proposes and the tool itself.

General Remarks

1. Throughout the manuscript the authors make a clear distinction between data generated by COSMOS and data generated by CARNIVAL. This distinction implies that these tools are separate and limits COSMOS as a tool to analyze differential expression. The addition of metabolomics in trans-omics research is certainly useful, but COSMOS may be best utilized as part of Omnipath. Furthermore, the flexibility and applicability of COSMOS to other datasets is unclear.

2. Extensive prior work with ccRCC has highlighted the importance of changes in the TCA cycle in ccRCC tumors. It is discouraging that COSMOS network did not capture this. The authors note

support for their hypotheses on a gene by gene basis but this work would benefit from comparison of their network to pathways identified in other trans-omic studies of ccRCC (citations below). Furthermore, the authors do not provide any biological evidence supporting the validity of mechanistic hypotheses generated by their network. This evidence would demonstrate the utility of COSMOS and would build upon previous trans-omics studies related to ccRCC.

Hakimi AA, Reznik E, Lee CH, et al. An Integrated Metabolic Atlas of Clear Cell Renal Cell Carcinoma. *Cancer Cell*. 2016;29(1):104-116. doi:10.1016/j.ccell.2015.12.004

Popławski P, Tohge T, Bogusławska J, et al. Integrated transcriptomic and metabolomic analysis shows that disturbances in metabolism of tumor cells contribute to poor survival of RCC patients. *Biochim Biophys Acta Mol Basis Dis*. 2017;1863(3):744-752. doi:10.1016/j.bbadis.2016.12.011

Cancer Genome Atlas Research Network. Comprehensive molecular characterization of clear cell renal cell carcinoma. *Nature*. 2013;499(7456):43-49. doi:10.1038/nature12222

Clark DJ, Dhanasekaran SM, Petralia F, et al. Integrated Proteogenomic Characterization of Clear Cell Renal Cell Carcinoma. *Cell*. 2020;180(1):207. doi:10.1016/j.cell.2019.12.026

3. The authors recognized the importance of validating the COSMOS network. However, their validation seems circular in that they use information that was excluded from their final network, but was considered as input for the network. This is an important problem that needs to be addressed. To test the consistency of the network we propose a 5-fold validation method where the authors divide their input node data into 5 groups and withhold one group and generate the network with the remainder. A true positive rate could be calculated by examining a pair of nodes that the network predicts would interact and determining whether the observed effect directions are consistent with what the network predicts. Similarly, a true negative can be calculated by examining cases where the network correctly does not place a connection between a pair of nodes in the withheld group. In addition, validation could be performed by using an independent dataset. A dataset was published in Clark et al. *Cell* 2019 (full citation above) with transcriptomic and phosphoproteomic data for 110 ccRCC patients, though this study does not include metabolomics, it would be interesting to see the network your tool develops using their dataset and may emphasize the value of metabolomics in ccRCC.

4. This manuscript would benefit from additional transparency in the methods and by supplying additional information as supplementary tables, see minor remarks.

Minor Remarks:

1. Methods

- Explain number of technical replicates for each data type and how they were processed in batches/blinded. We suggest a table explaining which samples were used for which data type, how many technical or biological replicates for each sample and a diagram of the processing pipeline. The phosphoproteomics section is well detailed, however, the TMT labelling information was confusing as it was not clear how many replicates of which samples were being used.
- The account for ProteomeXchange did not work.
- Given the paired data structure of your dataset, did you implement a paired analysis in LIMMA?
- In 4.5 the authors comment that they chose cutoffs for their omics data such the input data was a "comfortable size". Is this in reference to computational time, interpretation of the network? How sensitive is the network to changes in these cutoffs?
- In describing the Meta PKN contextualization the authors say that, "there were no incoherences in

the predicted activity signs between the common part of the two resulting networks, they were simply merged together, resulting in a combined network of 250 unique edges". Further elaboration on how the networks are combined is necessary. Is this process automated or manual? What factors are considered during this merging process?

2. Results

- Missing Supplemental Tables - Supp Table 1 should also outline which samples were used for which analyses as they change for each data type
- Provide supplemental table(s) with data on the 11 phosphosites and 21 metabolites used in network model. How many unique proteins are represented in the 11 phosphosites? Can the authors account for discrepancies in their expression and phosphosite data compared to previous trans-omics research on ccRCC?
- Authors should address the fact that their network is largely based on transcript level data, can they highlight examples where the phosphosite and/or metabolite data were critical in identifying a novel hypothesis?
- Are the transcription factors you detect as dysfunctional tissue specific? It would be interesting to know if you are picking up on some kidney-specific gene regulation.
- Provide citations for the following statement: "For instance, hypoxia, inflammation and oncogenic markers were up-regulated in tumors compared to healthy tissues".
- The authors use reference 23 to support the statement: "...among suppressed TFs we identified, HNF4A has been previously associated with ccRCC". This is accurate, however, the authors neglect to mention reference 23 reported that HNF4a is frequently reduced in renal cell carcinoma whereas Figure 2A suggests it is increased in data presented here.
- Figure 2A: The x-axis labels should be lined up more directly with the bars to facilitate interpretation. Further, it would be valuable to group the proteins by class, i.e. cluster the TF, the kinases, and the phosphatases.
- Figure 2C: The right panel shows a single blue dot in the top 10 targets, yet the left panel does not have a blue dot.
- Figure 4: Recommend black edges for visibility. Also, counter-intuitive that negative regulation/inhibition uses a pointed arrow while positive regulation/activation uses a flat arrow. Examination of the colors in the network and the arrows suggest that perhaps your legend is backwards for the Arrow Shape Effect?
- Figure 4: For a kinase, can you make the distinction between increased activity as measured by differential expression as opposed to increased activity as measured by phosphosite enrichment? Similarly, does a transcription factor have increased activity because expression levels of the transcription factor itself are increased or because the targets of the transcription factor are increased?

3. Discussion:

"It also predicted a depletion of adenine and consequently the down-regulation of PDPK1 activity through CXCR4." Adenine appears to be an input node in Figure 4A, so the decrease is observed not predicted? Careful discussion of what is actually observed as opposed to what the model predicts is necessary.

4. General:

- By specifically referencing either COSMOS or CARNIVAL the authors imply that they are separate tools. It would be more useful if CARNIVAL were integrated into COSMOS for utility and also in the manuscript as switching between the two names throughout the manuscript is challenging. Additionally, it is unclear the output of COSMOS/CARNIVAL will be for future users. It would be valuable to provide documentation on using the tool as part of the manuscript review.

- Throughout the manuscript authors refer to transcription factor and kinase activity as measured by transcript expression. Activity is a poor word choice as it specifically refers to catalytic activity. The phosphosite data demonstrates an enrichment of a particular substrate phosphorylation event, not necessarily increased general kinase activity. Similarly, increased expression of genes targeted by a particular transcription factor does not mean that this particular transcription factor has increased activity - transcription factors do not have catalytic activity.
- Throughout the manuscript gene names are not italicized.
- Throughout the manuscript numbers in the thousands do not have commas. ex. 32586 instead of 32,586
- Throughout the manuscript there are inconsistencies in how tool names are reported. ex. limma vs. LIMMA, Omnipath vs. OmniPath, DOROTHEA vs. DoRothEA

Reviewer #1:

In this manuscript, Dugourd et al. presents a method named COSMOS that integrates transcriptome, phosphoproteome and metabolome data into a directed network representing prior knowledge on transcription regulation, signaling transduction via phosphorylation, and cellular metabolism. The authors applied this method to a multi-omic dataset from a total of 11 renal cell carcinoma patients, and demonstrated that COSMOS was able to generate meaningful and reasonable biological hypotheses. The method seems to be a natural extended application of the CARNIVAL algorithm that the authors have developed previously to multi-omic data and multi-omic prior knowledge network (PKN). However, the authors has pointed out that this methodological framework can in theory be applied to any types of multi-omic data where the corresponding PKN can be reliably constructed, which makes COSMOS interesting for potentially broader applications. Overall we feel that the manuscript is clearly written and the results quite interesting. Our specific comments are as follows.

We appreciate the overall positive comments on our work and provide point by point responses to the questions below.

Major comments:

1. One key novelty of the COSMOS method over the previous CARNIVAL method is that COSMOS extends the application to multi-omic data. Therefore it can be important and valuable to show that the added domains of omic data can really lead to improved performance (e.g. accuracy of the predictions) over a single omic approach. The rationale of this comment is that due to technical limitations and noise in data, different omic data can have very poor correlation or consistency, and adding further omic data may not lead to a significant gain in practise.

We agree with this rationale, which is something that we were carefully considering when developing COSMOS. To minimise the lack of consistency between different omics data, we worked with a dataset where each omics layer was measured from the same snap frozen tissue sample. This point is now made clearer in the new Appendix Figure S9. We agree that technical limitations can affect the amount of information that can be extracted from combining together these different layers, as we explore below.

Appendix Figure S9

Schematic of the process sample collection and multi-omics data generation.

Specifically, for example, comparing the COSMOS prediction using the full data and PKN (involving three types of omic data) to that using only two of the three types of omic data, does the former have higher true-positive rate as measured from the consistency analysis the authors used?

We agree that it is relevant to show how COSMOS performs when increasing the number of omics layers that are integrated. Thus, we ran COSMOS (i) with all three layers (TFs, kinases/phosphatases and metabolomics data), (ii) with only TFs and metabolomics data, and (iii) with only kinases/phosphatases and metabolomics data. We compared the outputs of each run and performed the correlation network consistency analysis for each of them. We describe the results in the following paragraph that was added to the manuscript:

“Assuming thresholds of absolute values of correlation ranging between 0 and 1 to define true positive co-regulations, the comparison between the topology driven coregulation network and the data driven correlation network yielded a TPR ranging between 0.55 and 0 (n = 269 pairs of predicted/measured co-regulations) for the predictions (Appendix Figure S6). It performed consistently better than a random baseline (see methods) over the considered range of correlation coefficient thresholds. We also compared the results with network solutions obtained hiding either TFs or kinases/phosphatases. When TFs were hidden, COSMOS performed consistently better than the random baseline, and reached a maximum TPR of 0.62. Of note, this curve was estimated from only n = 21 coregulation events. When kinases and phosphatases were hidden, COSMOS performed again consistently better than the random baseline, and reached a maximum TPR of 0.58 (n = 228). In both cases the performance of COSMOS was slightly larger than the full COSMOS performance (TPR = 0.55). This could be due to a lack of consistency across the omics data, although due to the low number of comparisons we could not make a conclusive statement. These results suggest that COSMOS’ performance is relatively robust to removing either the phosphoproteomics or transcriptomics layers when trying to find connection between signaling and metabolism. However, using the three omics layers together yielded a larger network (367 edges (full) (full) vs 294 (hidden kinases) (hidden kinases) and 135 edges (hidden TFs)) and denser (1.67 edge/node ratio vs 1.54 and 1.19 edge/node ratio, respectively) than when one omics layer was removed (Dataset EV3). Hence, considering all layers yield a greater number of mechanistic hypotheses, even if not necessarily of higher quality.”

These results highlight that integrating both kinases/phosphatases and TFs when searching for regulation events between signaling and metabolism will lead to a larger and broader amount of mechanistic hypotheses to explore. At the same time, our results indicate that including more omics layers does not necessarily improve the quality of the predictions, along the lines of the reviewers comment above.

Appendix Figure S6

Comparison of COSMOS network co-regulation predictions with data-driven co-regulations between kinases phosphatases and TFs. Top panel shows the performance of COSMOS with all three omics layers. Middle panel shows performance when TFs are hidden. Bottom panel shows performance when TFs are hidden. Each panel compares the ability of COSMOS to capture co-regulation events between kinases/phosphatases and transcription factors that are consistent with observed correlations in the data.

	Number of inputs	Number of edges	Number of nodes	Edge/node ratio
Three layers	118	367	220	1,67
Kinases/phosphatases and metabolites	25	135	113	1,19
TFs and metabolites	93	294	190	1,54

Appendix Table 3

2. One key novelty and potential value of COSMOS (and CARNIVAL) over common pathway enrichment or the various "footprint-based" methods is that COSMOS can infer "causal" links that represent hypotheses about the potential mechanisms of actions underlying the observed biology, which can be further investigated using experiments. We understand that additional experimental validation may be out of the scope of this study, but we think that it is highly desirable if the authors can try to provide more direct validations of their inferred causal "paths", beyond the consistency analysis the authors have shown. This may be achieved using existent data. For example, it can be possible to mine public datasets where an input node or "upper-stream" node gene is knocked out or inhibited with drugs, and demonstrate with such data that some of the downstream node changed in the direction as predicted by COSMOS.

We agree that it is important to assess the ability of our approach to generate hypotheses that are consistent with data. In the CARNIVAL publication (Liu et al. 2019), we studied the consistency of CARNIVAL predictions with experimental perturbations, and found that causal pathway inference with CARNIVAL - the backbone of COSMOS - outperforms classic pathway analysis approaches such as GSEA. To expand these comparisons to COSMOS, we would need similar perturbation datasets with transcriptomics, phosphoproteomics and metabolomics, that we are not aware of.

As an alternative way to evaluate the relevance of COSMOS results, we have mined the literature to see if the mechanistic hypotheses obtained with COSMOS are coherent with what has already been reported in the context of ccRCC. We focused mainly on the interferon gamma response because it was enriched in our COSMOS network solution network. COSMOS highlighted many relevant crosstalks, some of them being actually drug targets in ccRCC therapies. They are outlined in the following paragraph that was added to the manuscript result section:

"We then used our network to investigate the regulation of relevant signalling cascades and metabolic reactions in ccRCC. An over-representation analysis of the network solution nodes (with the hallmark genesets of MSigDB) displayed the interferon gamma (IFNg) response as the top significant pathway in our COSMOS network. Hence, we focused on the interaction members of this pathway (such as NFKB1, HIF1A and PNP) and their crosstalks with metabolic deregulations to assess the relevance of the mechanistic hypotheses generated by COSMOS. We found that NFKB1, a central actor of the IFNg pathway is activated in ccRCC, consistently with other reports (Zhang et al, 2018; Rodrigues et al, 2018) where it was also demonstrated to be regulated by the PI3K/AKT pathway (An & Rettig, 2007). Interestingly, COSMOS also proposed the activation of BCAT1, one of the key enzymes of the branched-chain amino acid metabolism, orchestrated by HIF1A and MYC (Gordan et al, 2008; Ananieva & Wilkinson, 2018). Both mechanisms are shown in Figure 4 (1) and (2).

Of note, COSMOS provided deeper insights into these molecular mechanisms by linking MYC activation to NFKB1. The COSMOS model suggests that MYC upregulates the expression of the metabolic enzyme BCAT1, potentially leading to the observed higher levels of glutamate, glutamine and reduced glutathione in ccRCC (marked as (2) in Figure 4). A strong role of MYC and glutamine metabolism in ccRCC development is known (Shroff et al, 2015). Consistently with what was hypothesised in a recent proteogenomics ccRCC study (Clark et al, 2020), we were able to capture crosstalks between members of the interferon gamma pathway (such as JUN), YY1 and metabolic down-regulation observed in our data ((3) in Figure 4). COSMOS highlighted how YY1 inhibition might be connected with the depletion of adenine, hypoxanthine and inosine through regulation of the ADA and PNP metabolic enzyme (Popławski et al, 2017). The low levels of adenosine predicted by COSMOS might also be potentially linked to the down-regulation of AKT3 and up-regulation of YES1, through a cascade which involves both ADORA2B and GNAI1, downstream of s-Adenosyl-L-homocysteine and inhibition of KMT2A ((4) in Figure 4). Finally, COSMOS model shows a significant activation of MAPK1 and SMAD4 downstream of YES1 (a member of the SRC family) ((5) in Figure 4).

We also added this paragraph to the discussion section:

“Since the interferon gamma response pathway was the most over-represented cancer hallmark in the COSMOS network solution, we investigated further the relevance of the mechanistic hypothesis connecting members of this pathway. The network showed that the crosstalks between MAPK1, NFKB1, MYC, HIF1A and YY1 could explain the deregulation in glutamine and reduced glutathione metabolism, as well as inosine, hypoxanthine and adenine. These were particularly relevant as they were important interactions in ccRCC. MYC and glutamine metabolism appear to be an interesting therapeutic target of ccRCC (Shroff et al, 2015). YY1 is a known indirect inhibitor of MYC involved in cancer development (Austen et al, 1998). The COSMOS network showed YY1 could also potentially have a role in the down-regulation of the ADA and PNP metabolic enzyme activities. Coherently, PNP has been shown to be non-essential in ccRCC cell lines, which is expected from down-regulated metabolic enzymes (Gatto et al, 2015). In addition, the link shown by COSMOS between NFKB1 and MYC can have implications for the treatment of ccRCC, due to its pivotal role in arsenite (a drug used in chemotherapy) treatment of cancer (Huang et al, 2014). Furthermore, the activation of the NFKB1-MYC link in FBW7 deficient cells seems to sensitise them to Sorafinib (a MEK-Raf inhibitor), a drug used in treatment of primary kidney cancer (Huang et al, 2014). In addition, NFKB1 and MYC are both promising ccRCC treatment targets (Peri et al, 2013; Bailey et al, 2017). The link shown by COSMOS between KMT2A and adenosine is interesting, because KMT2A mutations have been reported in a number of ccRCC patients (Yan et al, 2019), suggesting that this enzyme might play a functional role in ccRCC development. Moreover, it has been proposed, at least in vitro, that ccRCC cell lines with low basal levels of phospho-AKT were sensitive to treatment with an adenosine analog (Kearney et al, 2015). The link between YES1, MAPK1 and SMAD4 in the COSMOS network is especially relevant considering that YES1 is a known targetable oncogene (Hamanaka et al, 2019). These examples illustrate the ability of COSMOS to extract mechanistic hypotheses to understand and potentially improve treatment of cancer by integration of multiple omics data and prior knowledge.”

Figure 4 - COSMOS subnetwork centred on the interferon gamma response pathway.

We extracted the main members of the interferon gamma response pathway (the most enriched cancer hallmark in the full COSMOS network). We also display here the metabolic enzymes that were hypothesised to be influenced downstream of this pathway, such as BCAT1 and PNP.

Additionally, we also ran COSMOS on the ccRCC CPTAC patient cohort dataset, as detailed in response to comment number 3 of reviewer #3.

3. The authors mentioned that the method relies on reliable sources of prior knowledge to construct a decent PKN. Therefore the important question is how robust the predictions are subject to the quality of the PKN? It may be desirable to perform a robustness analysis by, e.g. removing a certain fraction of the edges from the PKN and assess how much the results are affected.

We fully agree with this remark. We performed a set of simulations to analyze the robustness of COSMOS networks with regards to changes in the PKN.

We generated 6 alternative versions of the original PKN by randomly shuffling 2%, 10%, 20%, 30%, 40%, and 50% of the edges. Then, we compared the edges of the resulting network solutions. We computed the absolute difference of the edge weights (an edge weight being the frequency of appearance of a given edge in the pool of solutions reported by CPLEX, between 0 and 100%) between the original COSMOS network solution and each shuffled COSMOS network solution. We first looked at the “forward” networks (signaling -> metabolism). The results show that when 2% of edges are shuffled, the median weight difference is relatively low (median = 10 %, meaning that for 50% of the edges, the weight

difference was 10 % or less, with 4 % of edges changing from 0 to 100 weight or vice-versa), and it increases when the percentage of shuffled edge increases (median = 35% for 50% of edge shuffled). We then looked at the “backward” networks (metabolism -> signaling). Here the results were much less quantitative because only one solution was returned by CPLEX instead of a family of solutions. Thus, the weight differences could only be estimated as 0 or 100. Nevertheless, more than half of the edges of the backward network were common between the original and 2% shuffled backward COSMOS network solution. In summary, all resulting networks were of similar size, and largely overlapping, indicating that even for heavily distorted PKN (50% edges randomly shuffled), the underlying CPLEX formulation wasn't perturbed. We have added the following paragraph to the results section:

“To study the robustness of COSMOS to changes in the PKN, we generated a series of partially degraded PKN by randomly shuffling an increasing number of edges in the original PKN (2, 10, 20, 30, 40, 50% of all edges shuffled completely randomly). We ran COSMOS with each version of the PKN. We first compared the results of the “forward” COSMOS runs (connecting TFs and kinases with downstream metabolites). We calculated the absolute difference between the edge weight of the results (see Material and Methods, Meta PKN contextualisation) obtained from each shuffled PKN with the result obtained from the original PKN. The edge weight represents the frequency of appearance (in %) of an edge across all the networks in the pool of network solutions. This showed that for the 2% shuffled network, the differences were relatively small (median of the absolute weight difference = 10), with 4 % of edges flipped (i.e., 0 weight in shuffled network and 100 weight in original network, or vice-versa). As expected, the differences were higher regarding the other shuffled networks, with medians weight differences of 10, 14, 23, 50, 28 and 35 for the 2%, 10%, 20%, 30%, 40% and 50% shuffled PKN, respectively (Appendix Figure S7A).

We then compared the results of the “backward” COSMOS runs (connecting metabolites with downstream TFs and kinases). Here the comparison was far less quantitative because the optimization reported only a single solution for all runs except in the case of the 20% shuffled PKN (Appendix Figure S7B). 61%, 31%, 18%, 12%, 8% and 8% of edge weight differences were equal to 0 for the 2%, 10%, 20%, 30%, 40% and 50% shuffled PKN, respectively.

In both “forward” and “backward” runs, the network results had a relatively similar number of edges from the original and shuffled PKNs (min = 142, max = 342, mean = 263, sd = 63). The optimization thus consistently excluded a common set of edges covering the vast majority of the network, that contains over 56,000 edges.”

Forward COSMOS weight comparison

Backward COSMOS weight comparison

Appendix Figure 7

Distribution of edge weight differences between A) 'forward' and B) 'backward' results obtained from the original PKN and 2, 10, 20, 30, 40 and 50% shuffled PKNs. Each dot represents the absolute weight difference for a given edge. The diamonds represent the medians of the weight difference distributions. The boxes cover 25th to 75th percentiles of the distributions.

4. In the metabolomic part of the PKN construction, when a metabolic enzyme X catalyzed the conversion of metabolite A to metabolite B, the current network structure representing this is "A  X  B". However, another equally reasonable "causal" path is "B --| X --| A" (i.e. the product B inhibits the enzyme X in a negative feedback via allosteric regulation; and X consumes the reactant A, therefore effectively represented by an inhibition edge). Both paths may be added to the PKN without additional prior knowledge on which actually happens.

We thank the reviewer for this suggestion. Allosteric regulations are already included on COSMOS PKN if they are reported with high confidence in the STITCH database (these are general metabolite/protein inhibition, not necessarily between enzymes and their products). We also find that the idea of metabolic consumption inhibitory link is a good idea and we plan to incorporate it in follow-up studies. Such a question deserves a study of its own. The consumption/production balance between reactant and product could be fully modeled in the causal format of the prior-knowledge network in theory. However, it would also require thorough testing and refinement to optimize it when solving such networks with COSMOS. As a first step, we feel that it is already informative to model only the production of metabolites in the causal network. It allowed us to build a relatively intuitive problem to solve and interpret, as we only have to search for which up-stream regulation event may affect the production of given metabolites.

Regarding this comment, we have added the following statement in the discussion:

“Related to the importance of prior knowledge, the PKN can also depend on how we interpret the information we have about molecular interactions. In particular, we converted the reaction network of Recon3D into a causal network where metabolite reactants “activate” metabolic enzymes, and metabolic enzymes “activate” metabolite products. This first approximation assumes that metabolite abundances are only driven by their production rates. We plan to refine this in the future to include that metabolite abundances can change as a result of consumption as well.”

5. In the section, 'Building the multi-omics dataset', the authors concluded that TF dysregulation is more pervasive than signaling, transcription, and metabolism. Firstly, it is not explained well how the author came to the conclusion.

Secondly, it may help strengthen this claim by a plot of fold change (or Effect Size) for each gene for all the transcripts, metabolites, and phosphites.

We thank the referee for raising this point, and giving us the possibility to clarify this part of the manuscript. We relied on the PCA plots to affirm that there is a greater variance of gene expression explained by the conditions difference for transcriptomics compared to phosphoproteomics. We agree that more evidence can and should be provided to conclude that the transcriptional layer was more pervasive. As advised, we have made a volcano plot

(Appendix Figure S2), which supports the fact that the fold changes observed at the RNA sequencing level reach larger magnitudes than the ones from the phosphoproteomic data.

We replaced that sentence in the manuscript with :

“Consistently with the PCA, a volcano plot overlapping the results of the differential analysis of each omics showed that the transcriptomics dataset led to larger differences and smaller p-values than phospho-proteomics and metabolomics extracted from the same samples (Appendix Figure S2).”

Appendix Figure S2

Combined volcano plot showing the magnitude (fold-change, x-axis) and significance (P value, inverted y-axis) of the differential analysis of RNAseq, phosphoproteomics and metabolomics datasets.

Minor comments:

1. Does the PKN contain cycles (i.e. loops) that are inconsistent within itself (i.e. contain inconsistent signs)? If yes, does such structures pose a problem in the computation or interpretation of results, and how are they handled in the method? If not, are these structures removed explicitly and how?

Loops are not removed explicitly from the PKN. The ILP formulation connects the lower layer (metabolites in the forward run, TF/kinases in the backward run) to the top layer (TF/kinases in the forward run, metabolites in the backward run) by searching step by step possible “sign coherent upstream nodes” for the current layer (starting with the lowest layer). Thus, negative feedback loops will be automatically discarded as they will always lead to a sign incoherence when climbing to the next layer. Positive feedback loops will be sign coherent but will not help to further reach the top layer, so they will be ignored as they just increase the number of edges of the solution network without contributing to the overall fit. There is an implementation of CARNIVAL for data with multiple time points that can recover feedback loops, but we will require an appropriate multi-omics/multi-time-point dataset to test it in the context of COSMOS. However, loops can still be found in the final merged COSMOS network solution, when nodes are overlapping between “forward” and “backward” runs. To clarify this, we added the following part in the result section :

“CARNIVAL finds a direct path connecting downstream measurements with upstream nodes, and thus the solution networks do not contain loops. Loops can however appear in the final merged network when nodes are overlapping between “forward” and “backward” runs.”

2. We feel that the Methods need to be expanded to provide additional necessary details. For example, how was the integer programming (ILP) part of the COSMOS algorithm formulized? It may help to explicitly describe the ILP formulation of the problem. The part on the "footprint-based" inference of transcription factor (TF)/kinase/enzyme activity also needs further clarification -- to readers not familiar with such methods, it is not clear how this works or what algorithm(s) is/are used for TF/kinase/enzyme.

We have further expanded the text to clarify these points. We have added the following paragraphs in the results section:

“Footprint-based activity estimation(Dugourd and Saez-Rodriguez 2019) relies on the concept that the measured abundances of molecules (such as phosphopeptides or transcripts) can be used as a proxy of up-stream (direct or indirect) regulator activities responsible for those changes(Casado et al. 2013; Ochoa et al. 2016; Rhodes et al. 2005). In the case of TF activity estimation, this means that measured changes in the abundances of transcripts give us information about the changes of activities of the transcription factors that regulate their abundance. An activity estimation only depends on the changes of the abundances measured in its target transcripts, not its own transcript abundance. In this study, we used the VIPER algorithm(Alvarez et al. 2016) to estimate the activity of transcription factors and kinases based on transcript and phosphopeptide abundances changes, respectively.”

“COSMOS uses the CARNIVAL R package to perform the network optimization via an ILP algorithm. In brief, we try to minimise the value of an objective function that depends on two main factors: 1) the mismatch between the simulated values of kinases, TFs and metabolites for a given causal network and the corresponding available values estimated from the measurements and 2) the size of the solution network. For each run, given the prior knowledge network and the input and measurements, a set of constraints are generated to define the solution space (based on the objective function) that the ILP solver (IBM CPLEX in

our study) explores to find an optimal solution(Liu et al. 2019; Melas et al. 2015). After a given amount of time (decided by the user), the search is stopped and the best solution at this point is returned by CPLEX. The solution is usually in the form of a pool of networks that are all equally optimal, with respect to the objective function. Thus CARNIVAL reports the solution as a set of edges with an associated weight that represent their frequency of appearance in the current network pool.”

3. In Figure 4, the "Edge arrow shape" and the corresponding "Effect" legend seems to be wrong?

We apologize for this error and have corrected it accordingly.

Reviewer #3:

Summary

The authors developed COSMOS, a novel Prior Knowledge Network (PKN), that uses a previously published tool, CARNIVAL, to generate networks using transcriptomics, metabolomics, and phosphoproteomics data. They applied this tool to a dataset of tumor and normal tissue samples from 11 ccRCC patients. Their work identifies transcription factors, kinases, and metabolites that are dysregulated in ccRCC and, using COSMOS/CARNIVAL, generate a network that suggests novel mechanistic hypotheses. The authors have generated a useful PKN and tool for analyzing a novel combination of omics data. However, the utility of COSMOS/CARNIVAL as a tool for generally combining multiple omics datasets is unclear. This study may contribute novel understanding of ccRCC biology, however, further validation of their network is necessary to validate both the hypotheses their network proposes and the tool itself.

General Remarks

1. Throughout the manuscript the authors make a clear distinction between data generated by COSMOS and data generated by CARNIVAL. This distinction implies that these tools are disparate and limits COSMOS as a tool to analyze differential expression.

We apologize for this confusion. COSMOS uses the underlying ILP optimisation strategy of CARNIVAL, but applies it in an expanded and more complex context than the original CARNIVAL paper. CARNIVAL aims at connecting TF activity estimations to potential upstream signaling perturbations with causal links. COSMOS aims at searching for potential causal links *between* measurements (such as metabolites) and activity estimations (such as TF and kinase activities, instead of known perturbations). In addition, COSMOS does so in a more general manner (by allowing to use footprint-based activity estimations instead of experimental perturbations), allowing for more types of omic as inputs.

Also of note, COSMOS can use a different approach than CARNIVAL to perform the network optimization.

To summarise, COSMOS generalises the underlying concept of CARNIVAL's causal network optimisation to systematically integrate data from multiple omics layers with prior-knowledge. In the future, we will expand the COSMOS R package to include more prior knowledge resources and more causal network optimisation algorithms.

To make the role of COSMOS clearer to readers, we now directly refer to COSMOS instead of CARNIVAL when talking about the final solution network generated by COSMOS. For example, CARNIVAL was replaced by COSMOS in the following text :

“One way to estimate the performance is to check if the COSMOS mechanistic hypotheses correspond to correlations observed in tumor tissues (Appendix Figure S3B). Thus, on the one hand, a topology driven coregulation network was generated from the COSMOS network.”

Furthermore, we have added this paragraph to make the role of CARNIVAL in the COSMOS method clearer:

“The ILP algorithm that COSMOS relies on is handled by the CARNIVAL R package. In brief, CARNIVAL tries to minimise the value of an objective function that depends on two main factors : 1) the mismatch between the simulated values of a given causal network and activities of selected input nodes (upstream regulators) and the corresponding available downstream measurements and 2) the size of the solution network. For each CARNIVAL run, given the prior knowledge network and the input and measurements, a set of constraints are generated to define the solution space (based on the objective function) that the solver (IBM CPLEX in our study) will explore to find an optimal solution (Liu et al, 2019a; Melas et al, 2015). After a given amount of time (decided by the user), the search is stopped and the best solution at this point is returned by CPLEX. The solution is usually in the form of a pool of networks that are all equally optimal, with respect to the objective function. Thus CARNIVAL reports the solution as a set of edges with an associated weight that represent their frequency of appearance in the current network pool.”

Finally, we have developed an R package for COSMOS available at: <https://github.com/saezlab/COSMOS>

The addition of metabolomics in trans-omics research is certainly useful, but COSMOS may be best utilized as part of Omnipath.

Indeed we have used COSMOS downstream from Omnipath, using prior knowledge on signaling and gene-regulatory processes from Omnipath. COSMOS also uses complementary knowledge from other resources: Recon3D and STITCH that provide information on metabolic interactions and metabolite-protein interactions, respectively. In addition, COSMOS is an analysis method that can run with other sources of prior knowledge, and it also provides the user with general network manipulation tools to lighten the load of the optimisation procedure. Hence it is an independent and synergistic tool from the meta-database Omnipath.

Furthermore, the flexibility and applicability of COSMOS to other datasets is unclear.

In order to demonstrate the flexibility of COSMOS to other datasets, we applied it on the ccRCC CPTAC dataset of Clark 2019 (as detailed in following answers) and to another breast cancer multi-omics dataset with transcriptomics and fluxomics.

The following paragraph was added the result section:

“We also compared the results we obtained from our samples with results obtained using another independent ccRCC dataset. We obtained the transcriptomics and phosphoproteomics dataset of the Clinical Proteomic Tumor Analysis Consortium (CPTAC) ccRCC patient cohort(Clark et al, 2020). Following the same approach as with our patient samples, we performed the differential analysis between tumor and healthy tissue for both omics datasets and estimated TFs and kinase/phosphatase activities. Then, we ran COSMOS to find mechanistic hypotheses explaining the connections between deregulated transcription factors and kinases/phosphatases. The resulting COSMOS network was coherent with the results shown in the original publication, and also provided additional

information on the crosstalks between deregulated kinases and transcription factors. In particular, COSMOS captured the signaling crosstalks between EGF, VEGF, AKT, MAPK, MTOR, NFkB and MYC (Appendix Table 4). Finally we compared which biological processes were captured in the COSMOS network generated from the data of our patient samples and the COSMOS network generated from the CPTAC ccRCC patient cohort. As shown in Appendix Figure S5, the top over-represented pathways were very consistent between the two studies. Notably, PI3K-AKT-MTOR signaling and G2M checkpoint(Clark et al, 2020), TNFA signaling via NFkB(AI-Lamki et al, 2010), interferon gamma response(Thapa et al, 2013), WNT beta catenin signaling(Xu et al, 2016), and IL6 JAK STAT3 signaling pathway were all significantly over-represented (p -value < 0.02).

Finally, we applied COSMOS to a public breast cancer dataset including transcriptomics and fluxomics measurements(Katzir et al, 2019) to connect signaling directly with metabolic flux estimation, instead of metabolite abundance measurements as done in the previous cases. We performed a differential analysis of transcript abundance and flux values between tumour cells cultured with and without glutamine. We then looked for mechanistic hypotheses connecting TF activity deregulations and changes in flux values. Coherently with the original study, almost all metabolites of the TCA cycle, glycolysis and pentose phosphate pathway were predicted to be down-regulated by COSMOS (Appendix Figure S8). Interestingly, COSMOS finds HIF1A as a master regulator of glycolysis through his effect on HK1/2, GAPDH, GCK, ENO1 and LDHA transcription. This is consistent with the known role of HIF1A in breast cancer (Masoud & Li, 2015; Zhang et al, 2015; Singh et al, 2017; Samanta et al, 2014). The down-regulation of MYC is also in line with the decreased activity of HK2 and LDHA and GLS1 enzymes which are important in aerobic glycolysis and glutamine catabolism(Dong et al, 2020)."

The following paragraph was added to the discussion section:

"We then applied COSMOS on two additional public datasets. The CPTAC ccRCC dataset with transcriptomics and phosphoproteomics showed that COSMOS could capture crosstalk in signaling pathways that were coherent with the findings of the original study. A breast cancer cell line dataset showed that COSMOS could also be used to find potential signaling mechanisms regulating reaction flux changes. In both cases, COSMOS hypotheses were overall coherent with the original study and additionally yielded mechanistic hypotheses of regulatory events."

Appendix Figure 5

Comparison of over-representation analysis performed with msigDB HALLMARK pathways between COSMOS networks generated from A) our patient samples and B) the CPTAC ccRCC patient cohort. Overall, a very similar set of pathways was significantly over-represented in both cases, notably PI3K-AKT-MTOR signaling, TNFA signaling via NFKB, interferon gamma response, WNT beta catenin signaling, G2M checkpoint and IL6 JAK STAT3 signaling.

Appendix Figure S8

COSMOS solution network connecting metabolic fluxes and TF activity deregulations observed in a breast cancer cell line cultured with and without glutamine(Katzir et al. 2019).

2. Extensive prior work with ccRCC has highlighted the importance of changes in the TCA cycle in ccRCC tumors. It is discouraging that COSMOS network did not capture this. The authors note support for their hypotheses on a gene by gene basis but this work would benefit from comparison of their network to pathways identified in other trans-omic studies of ccRCC (citations below). Furthermore, the authors do not provide any biological evidence

supporting the validity of mechanistic hypotheses generated by their network. This evidence would demonstrate the utility of COSMOS and would build upon previous trans-omics studies related to ccRCC.

This is an interesting point, which led us to further investigate deregulation of canonical pathways. It is widely accepted that TCA cycle and oxidative phosphorylation are targeted and deregulated in ccRCC (Hakimi et al. 2016). To understand why we didn't capture this alteration, we first sought whether evidence of TCA deregulation could be found at the level of individual omics datasets. Therefore, we first ran a pathway enrichment analysis on transcriptomics, phosphoproteomics and metabolomics data sets separately. This separated analysis showed that the oxidative phosphorylation hallmark pathway (OPHP) was actually significantly suppressed in our tumor cohort with respect to both phospho-proteomics and transcriptomics datasets, consistently with other studies. However, we could detect only a few significant depleted metabolites linked to OPHP (p-values < 0.05), namely succinate, isovaleryl carnitine and O-propionylcarnitine (malate, aconitate, citrate or fumarate were not significant with respect to our threshold). As a result, OPHP was only marginally enriched in the group of depleted metabolites in ccRCC tumours. This apparent discrepancy with the literature and the proteomics datasets is likely due to the resolution of our metabolomics dataset. Indeed the number of metabolites detected with LC-MS depends on several factors, including the type of chromatography column used, the ionisation of the molecules and their relative abundance. Finally, disruption and extraction of the tissue might represent another source of variability compared to different studies. Nevertheless, we believe that this potential technical limitation does not affect the overall validity of our data and model, since we were able to capture other metabolic pathways and signalling cascades associated with ccRCC.

Focusing on the metabolites linked to OPHP we detected, we can use COSMOS to generate mechanistic hypotheses relative to their regulation. For instance, COSMOS network actually shows that the down-regulation of these three metabolites is linked to upstream signaling events such as the deregulation of JUN, MAPK8/9 and RXRA (Reviewer figure 1, (1)) and inhibition of CREBBP and hydrogen peroxide (Reviewer figure 1, (2)). According to the model, the changes in these metabolites are mediated by alterations in the transcriptional regulation of metabolic enzymes. Thus, we checked if ACAA1 CPT1A and P4HA2 (i.e. the metabolic enzymes regulated by transcription factors leading to changes in the succinate, isovaleryl carnitine and O-propionylcarnitine abundances) transcript abundance are coherent with the model predictions. ACAA1 and CPT1A are regulated by the RXRA TF (Reviewer figure 1, (3,4)), while P4HA2 is regulated by the TP53 TF (Reviewer figure 1, (2)). ACAA1 transcript abundance is strongly down-regulated (limma t-value = -8.8) while CPT1A transcript abundance is marginally down-regulated (limma t-value = -1.8), supporting the COSMOS crosstalk hypothesis between MAPK, RXRA, CPT1A, ACAA1, propanoylcarnitine and isovaleryl carnitine (Reviewer figure 1, (1,3,4)). On the contrary, P4H2A transcript abundance is significantly up-regulated (limma t-value = 2.9), disproving a direct TP53-P4HA2-succinate regulation (Reviewer figure 1, (2)).

Reviewer figure 1

COSMOS mechanistic hypotheses to explain the measured depletion of oxidative phosphorylation related metabolites.

Finally, we would like to highlight that COSMOS manages to capture many relevant crosstalks of interest in ccRCC and among those, the interferon gamma response pathway. We chose to focus on this pathway in our manuscript because it was the most significantly overrepresented one in our COSMOS network solution. We detail this in our answer to the second comment of reviewer #1 (page 9-11).

3. The authors recognized the importance of validating the COSMOS network. However, their validation seems circular in that they use information that was excluded from their final network, but was considered as input for the network. This is an important problem that needs to be addressed.

We apologize if the consistency analysis wasn't explained clearly enough. First we would like to make it clear that this consistency analysis only aimed at exploring the agreement between information yielded by the integration of omics data with prior-knowledge and the original omics data itself. We know further validated COSMOS results by cross-checking them with independent studies (as detailed in our answer to the second comment of reviewer #1 (page 9-11)).

Regarding the consistency analysis, none of the information that is used were considered as inputs for COSMOS, as explained below. We originally did two consistency analyses.

In the first one, transcript abundance changes that are used to assess the coherence of COSMOS predictions are targets of TF that were *not* used as input for COSMOS. Only transcripts that are direct targets of COSMOS TFs inputs could have been considered as indirect inputs, and accordingly they were explicitly excluded from this consistency analysis method. In the new version of the COSMOS method, we are actually directly using the transcriptomics data to correct the output of the first CARNIVAL run. We decided to do it because this information was better used directly to correct COSMOS solutions. Thus, this part of the consistency analysis was removed.

To explain this, we added the following paragraph in the method section:

“Correction:

We use the transcriptomics data differential gene expression analysis results to directly remove any edge that leads to an incoherence between a TF activity and its target transcript abundance change (which is a wrongly predicted incoherent transcriptional regulation event). This is done once before running CARNIVAL, using TF activities predicted with DoRothEA. Then, we do a pre-run of CARNIVAL (TFs/kinases/phosphatases -> metabolites) to generate a first solution network. We can subsequently use TF activities predicted by CARNIVAL to filter out any wrongly predicted transcriptional regulation event from the meta PKN (Appendix Figure 3A).”

For the second consistency analysis, COSMOS de facto ignores any potential correlations between its inputs. It only uses the actual possible activity flows of the PKN to connect inputs together. This means that the coregulation network that can be derived from the COSMOS network relies on measurements and footprint-based activities that are considered independently from each other. They are connected together using only the interactions available in the PKN. Thus, the correlation between the measurements is not considered by COSMOS at any moment when building the network solution. Hence, we wanted to confirm that the topological information that COSMOS considered was consistent with correlations that can be derived from the single sample level TF and kinase activities, which do not use any of the topological prior knowledge that COSMOS relies on.

To test the consistency of the network we propose a 5-fold validation method where the authors divide their input node data into 5 groups and withhold one group and generate the network with the remainder. A true positive rate could be calculated by examining a pair of nodes that the network predicts would interact and determining whether the observed effect directions are consistent with what the network predicts. Similarly, a true negative can be calculated by examining cases where the network correctly does not place a connection between a pair of nodes in the withheld group.

To perform a cross-validation is a very valid suggestion for a predictive model, but, COSMOS aims at integrating measured data with prior-knowledge in a consistent and systematic manner, not to explicitly predict the outcome of new experiments. It provides the users with a set of coherent mechanistic hypotheses to explain changes observed in a given omics layer with upstream regulators from other omics layers.

In that sense, COSMOS would fall in the category of descriptive (or ‘unsupervised’) models, rather than predictive (‘supervised’) models. In the latter case, one predicts an output, and hence a cross-validation analysis as the reviewer suggests would be very valuable, to estimate the performance of models to make predictions out of their training set. However, there is no model training involved in the case of COSMOS, simply contextualisation of a generic PKN to a given set of inputs. This is done by finding which of the upstream nodes used as input can potentially explain downstream measurements in the most parsimonious manner.

To clarify this point we have added to the Discussion:

“Thus, we developed COSMOS to scale this type of reasoning up to the entire PKN with all significantly deregulated protein activities and metabolites. We relied on an ILP optimisation through the CARNIVAL R package(Liu et al, 2019b) to contextualise this PKN with our data. We refined the optimization procedure to handle this very large PKN, and built an R package to facilitate others to use it with their own data. Given a set of deregulated TFs, kinases/phosphatases or metabolites, COSMOS provides the users with a set of coherent mechanistic hypotheses to explain changes observed in a given omics layer with upstream regulators from other omics layers. Thus, its aim is to integrate measured data with prior-knowledge in a consistent and systematic manner, not to explicitly predict the outcome of new experiments.”

In addition, validation could be performed by using an independent dataset. A dataset was published in Clark et al. Cell 2019 (full citation above) with transcriptomic and phosphoproteomic data for 110 ccRCC patients, though this study does not include metabolomics, it would be interesting to see the network your tool develops using their dataset and may emphasize the value of metabolomics in ccRCC.

We agree that it is an interesting idea to try COSMOS on the CPTAC CCRCC dataset of Clark et al. 2019. We analyzed this data with COSMOS. Specifically, we tried to connect the kinase/phosphatase layer with the transcription factor layer, since both transcriptomics and phosphoproteomics were available, but not metabolomics. The results we obtained with this dataset were very coherent with the results reported in the publication and the results obtained from our patient samples. This is summarized in the following paragraph that we have added to the manuscript :

“We also compared the results we obtained from our samples with results obtained using another independent ccRCC dataset. We obtained the transcriptomics and phosphoproteomics dataset of the CPTAC ccRCC patient cohort. Following the same approach as with our patient samples, we performed the differential analysis between tumor and healthy tissue for both omics datasets and estimated TFs and kinase/phosphatase activities. Then, we ran COSMOS to find mechanistic hypotheses explaining the connections between deregulated transcription factors and kinases/phosphatases. The resulting COSMOS network was coherent with the results shown in the original publication, and also provided additional information on the crosstalks between deregulated kinases and transcription factors. In particular, COSMOS captured the signaling crosstalks between EGF, VEGF, AKT, MAPK, MTOR, NFKB and MYC (Appendix Table 4). Finally we compared which

biological processes were captured in the COSMOS network generated from the data of our patient samples and the COSMOS network generated from the CPTAC ccRCC patient cohort. As shown in Appendix Figure S5, the top over-represented pathways were very consistent between the two studies. Notably, PI3K-AKT-MTOR signaling and G2M checkpoint(Clark et al, 2020), TNFA signaling via NFKB(Al-Lamki et al, 2010), interferon gamma response(Thapa et al, 2013), WNT beta catenin signaling(Xu et al, 2016), and IL6 JAK STAT3 signaling pathway were all significantly over-represented (p -value < 0.02)”

4. This manuscript would benefit from additional transparency in the methods and by supplying additional information as supplementary tables, see minor remarks.

Minor Remarks:

1. Methods

- Explain number of technical replicates for each data type and how they were processed in batches/blinded. We suggest a table explaining which samples were used for which data type, how many technical or biological replicates for each sample and a diagram of the processing pipeline. The phosphoproteomics section is well detailed, however, the TMT labelling information was confusing as it was not clear how many replicates of which samples were being used.

We agree and we have added this information to the Appendix Table 1 and added the Appendix Figure S9.

We also added the following paragraph in the method section :

“We included a total of 22 samples from 11 renal cancer patients (6 men, age 65.0+/-14.31, 5 women, age 65.2+/-9.257(mean+/-SD)) for transcriptomics. Phosphoproteomics was also measured in a subset of 18 samples from 9 of these patients (6 men, age 65+/-14.31; 3 women, age 63.33+/-11.06(mean+/-SD)), and metabolomics was also measured in 16 samples from 8 out of these 9 patient (5 men, age 62+/-13.23; 3 women, age 63.33+/-9.89(mean+/-SD), Appendix Figure S9, Appendix table 1).”

- The account for ProteomeXchange did not work.

We are sorry to hear that the reviewer had problems with the account, and we have ensured that it worked. You can access it by going to <http://proteomecentral.proteomexchange.org/cgi/GetDataset>, use the keyword PXD018218 and follow the instructions for reviewers (credentials are Username: reviewer81921@ebi.ac.uk, Password: Kidney2020). The PRIDE database can be down from time to time, if you see the “500 Internal Server Error”, please try again at another time.

-Given the paired data structure of your dataset, did you implement a paired analysis in LIMMA?

Our tumor and healthy samples are indeed coming from the same patient. Due to this, it could appear natural to analyse them with a paired t-test or a mixed effect linear model. However, we made a PCA of our samples, and we didn't observe any noticeable patient effect for the metabolomics and phosphoproteomics data, while there was only a minor one apparent in the transcriptomics data (Appendix Figure 1). Importantly, we do not have actual pairs of metabolomics and phosphoproteomics measurements for many patients, due to missing values inherent to mass-spectrometry technology. Because of these reasons, we

originally settled for an unpaired differential analysis of our healthy and tumor sample. While patient specific effects may reduce our statistical power, we reasoned that it is preferable to the risk of including a poorly characterised patient specific effect in LIMMA's linear model.

Triggered by the reviewer's comment, we compared the result of our LIMMA analysis with the same analysis regressing out the patient specific effect with a linear model (effectively emulating a paired analysis). As expected, the results of both analyses are very similar (Reviewer figure 2). Yet, for many phosphosite and metabolites, the regression parameter corresponding to the patient specific effect is poorly estimated, due to the missing values of mass spectrometry data. For example, this is apparent in the case of quinolinic acid, where only one patient actually had a pair of measurements between healthy and tumor tissue. Coherently, the differences observed between the two analysis strategies are more pronounced in this case. Overall, the differences between the results of both analyses are mild.

With all this in mind, we conclude that it is not clear if a paired analysis is really suited to our study, and in any case we would expect very similar results.

Reviewer figure 2

Comparison of LIMMA analysis performed in an unpaired fashion and emulating a mixed linear effect model (paired analysis).

-In 4.5 the authors comment that they chose cutoffs for their omics data such the input data was a "comfortable size". Is this in reference to computational time, interpretation of the network?

The activity scores are enrichment scores that are z-transformed with an empirical null distribution. Thus, an absolute value of 1.7 would correspond to a p-value of 0.05. We also chose 0.05 as the p-value cutoff for the deregulated metabolites. The 0.05 value is a commonly used yet admittedly relatively arbitrary choice. In the absence of better evidence as to which threshold to choose, it seemed an appropriate decision - but as we elaborate below, we acknowledge its impact should be explored, and we have now done so.

To make this clearer, the following paragraph was modified in the method section:

"CARNIVAL needs a set of starting and end nodes to look for paths in between. TFs, kinases and phosphatases absolute normalised enrichment scores greater than 1.7 standard deviation were considered deregulated. Coherently, metabolites with uncorrected p-values smaller than 0.05 were considered deregulated. We give more information on the rationale to choose an appropriate threshold in the Appendix note 1. This yielded a set of 98 TFs, 25 kinases/phosphatase and 41 metabolites to be used as input and measurements for COSMOS."

How sensitive is the network to changes in these cutoffs?

We agree that the choice of cutoff can have an important impact on the resulting COSMOS network. In order to make it clearer how sensitive COSMOS can be to this cutoff, we detail its impact in a new Appendix note 1:

"The COSMOS network solution aims at connecting a downstream layer (the measurements) with upstream regulators (the inputs). The choice of the cutoff will determine which inputs and measurements will be connected together by COSMOS. Thus, if more (or less) inputs and measurements are provided, the network solution will contain additional (or fewer) edges to connect them."

Knowing that, the choice of the threshold has to be decided with respect to (i) which are the TFs, kinases, phosphatase and metabolites that a user wishes to potentially connect together, (ii) how confident the user is that the TFs, kinases, phosphatase and metabolites are actually deregulated.

To illustrate this and show how the COSMOS network may change with respect to cutoff changes, we made three additional runs of COSMOS (connecting downstream metabolites with upstream TFs and kinases, that is the "forward" run). We chose:

(1) a very loose cutoff (p-value < 0.5 and absolute activity score > 0.6 sd, essentially including everything),

(2) a cutoff reducing the number of upstream TFs and kinases used as upstream input while keeping the same measurements as the original COSMOS run (p -value < 0.05 and absolute activity score > 2.4 sd)

(3) a very stringent cutoff for both inputs and measurements (p -value < 0.001 and absolute activity score > 5.2 sd).

As expected, the loose threshold yielded the largest network (200 edges) while the most stringent one yielded the smallest (50 edges) compared the 162 edges of the original network. The second case yielded 108 edges. We compared the latter with the original network (in the same manner as for the network shuffling analysis, see Material and Methods, Meta PKN contextualisation for explanation on the edge weight) and found that the solutions were relatively similar, with a median absolute weight difference of 25%.

To conclude, the cutoff choice depends on the situation, and there will be - like is customary in such type of analysis - a tradeoff between coverage and reliability.

-In describing the Meta PKN contextualization the authors say that, "there were no incoherences in the predicted activity signs between the common part of the two resulting networks, they were simply merged together, resulting in a combined network of 250 unique edges". Further elaboration on how the networks are combined is necessary. Is this process automated or manual? What factors are considered during this merging process?

The two networks ("forward" and "backward") are merged by joining the interactions from both instances and dropping duplicate ones. Essentially, we are taking the union of the two sets of nodes and edges. The only factor that is effectively considered is whether or not there are common nodes between the two networks that have inconsistent signs. If there are common nodes between the two networks that have inconsistent signs, we include both predictions in the final merged network to denote the unknown variable concerning the actual sign of the node. This is in any case a rare event. In the result generated by the newer version of COSMOS, this was the case for the ARNT TF that was predicted with two different signs between the forward and backward run.

We modified the text accordingly in the method section to make this clearer :

"There was a single incoherence in the predicted sign of ARNT2 transcription factor (-1 in forward run, 1 in backward run) between the common part of the two resulting networks. We made the union of the two networks, resulting in a combined network of 449 unique edges, while preserving the incoherent sign of ARNT2 in the corresponding node attributes of the network (Appendix Table 5)."

2. Results

-Missing Supplemental Tables - Supp Table 1 should also outline which samples were used for which analyses as they change for each data type

We have added this information in Appendix Figure S1 (see Reviewer figure 3).

ID	samples	phospho	phospho TMT plex	RNA	metab	ischemia tim
11	11KI	yes	plex 3	yes	yes	0
11	11TU	yes	plex 3	yes	yes	0
15	15KI	yes	plex 1	yes	yes	0
15	15TU	yes	plex 1	yes	yes	0
16	16KI	yes	plex 1	yes	yes	0
16	16TU	yes	plex 1	yes	yes	0
24	24KI	yes	plex 3	yes	yes	19
24	24TU	yes	plex 3	yes	yes	19
29	29KI	yes	plex 1	yes	yes	15,5
29	29TU	yes	plex 1	yes	yes	15,5
31	31KI	-	-	yes	-	25
31	31TU	-	-	yes	-	25
32	32KI	yes	plex 2	yes	yes	0
32	32TU	yes	plex 2	yes	yes	0
35	35KI	yes	plex 2	yes	-	0
35	35TU	yes	plex 2	yes	-	0
36	36KI	-	-	yes	-	0
36	36TU	-	-	yes	-	0
38	38KI	yes	plex 4	yes	yes	0
38	38TU	yes	plex 4	yes	yes	0
40	40KI	yes	plex 3	yes	yes	12
40	40TU	yes	plex 3	yes	yes	12

Reviewer figure 3

Extra information indicating which patient had which type of data generated.

-Provide supplemental table(s) with data on the 11 phosphosites and 21 metabolites used in the network model. How many unique proteins are represented in the 11 phosphosites? Can the authors account for discrepancies in their expression and phosphosite data compared to previous trans-omics research on ccRCC?

We apologize for this misunderstanding but all phosphosites were used for kinase activity estimations, not just the 11 ones that are individually statistically significant. In order to avoid such potential misunderstanding, we modified the following section of the results accordingly:

“Each omics dataset was independently submitted to differential (tumor vs healthy tissue) analysis using LIMMA(Ritchie et al, 2015). Consistently with the PCA, a volcano plot overlapping the results of the differential analysis of each omics showed that the transcriptomics dataset led to larger differences and smaller p-values than phospho-proteomics and metabolomics extracted from the same samples (Appendix Figure 2). This is further apparent by the number of hits under a given FDR threshold. We obtained 6,699 transcripts and 21 metabolites significantly regulated with False Discovery Rate (FDR) < 0.05. While only 11 phosphosites were found under 0.05 FDR, 447 phosphosites had an FDR < 0.2. This result confirmed that tumor samples displayed molecular deregulations spanning across signaling, transcription, and metabolism. Then, the differential statistics for all tested (not just the ones under the FDR threshold) transcripts, phospho-peptides and metabolites were then used for further downstream analysis, as explained below.”

We have also added the following paragraph in the hope that our explanations on how an enrichment analysis is performed made this clearer too:

“Footprint-based activity estimation relies on the concept that measured omics abundances (such as phosphopeptides or transcripts) can be used as a proxy of up-stream (direct or indirect) regulator activities that are responsible for those changes (Casado et al, 2013; Ochoa et al, 2016; Rhodes et al, 2005). In the case of TF activity estimation, this translates into the fact that measured changes in the abundances of given transcripts give us information about the changes of activities of the transcription factors that regulates such abundances (Dugourd and Saez-Rodriguez 2019). This means that an activity estimation only depends on the changes of the abundances measured in its target transcripts, not its own transcript abundance. In this study, we use the VIPER algorithm (Alvarez et al. 2016) to estimate the activity of transcription factors and kinases based on transcript and phosphopeptide abundances changes, respectively.”

-Authors should address the fact that their network is largely based on transcript level data, can they highlight examples where the phosphosite and/or metabolite data were critical in identifying a novel hypothesis?

We apologise that the manuscript wasn't clear enough regarding this point: 98 TF, 25 kinases/phosphatase and 41 metabolites are used as input to generate the solution network. The TF and kinase/phosphatases are estimated from the entire transcriptomics and phosphoproteomics data set, as explained above. While there are more TFs than kinases and metabolites, their number doesn't appear overwhelming.

Finally, every hypothesis we generate in the paper links kinases, TFs and metabolites, hence using all available omic data.

-Are the transcription factors you detect as dysfunctional tissue specific? It would be interesting to know if you are picking up on some kidney-specific gene regulation.

We looked if COSMOS was able to capture mechanisms related to the HNF transcription factor family, which regulates kidney specific transcriptional programs. The TF activity estimation highlighted a downregulation of HNF4A and HNF1B. While they were both used as input for the COSMOS network generation, the algorithm only included HNF4A in the final optimal COSMOS network solution. Interestingly, COSMOS predicts that PRKAA1 is inhibiting HNF4A activity.

-Provide citations for the following statement: "For instance, hypoxia, inflammation and oncogenic markers were up-regulated in tumors compared to healthy tissues".

We now provide citations for this statement: hypoxia (Schödel et al. 2016), inflammation (Zeng et al. 2014), oncogenic markers (Clark et al. 2020). This is now apparent in the text :

“For instance, hypoxia (HIF1A), inflammation (STAT2) and oncogenic (MYC, Cyclin Dependent Kinase 2 and 7 (CDK2/7)) markers were up-regulated in tumors compared to healthy tissues (Figure 2) (Schödel et al, 2016; Zeng et al, 2014; Clark et al, 2020)”

-The authors use reference 23 to support the statement: "...among suppressed TFs we identified, HNF4A has been previously associated with ccRCC". This is accurate, however,

the authors neglect to mention reference 23 reported that HNF4a is frequently reduced in renal cell carcinoma whereas Figure 2A suggests it is increased in data presented here. Thank you for noticing this discrepancy. With the updated TF activities that were estimated with the dorothea package (a revised version of the regulons available in the omnipath web-service we used before), HNF4a is down-regulated. A few TFs such as this one and also TP53 where incorrectly signed when predicted using prior knowledge from the omnipath web-service.

-Figure 2A: The x-axis labels should be lined up more directly with the bars to facilitate interpretation. Further, it would be valuable to group the proteins by class, i.e. cluster the TF, the kinases, and the phosphatases.

We thank the reviewer for this suggestion. We updated the figure to increase the readability).

-Figure 2C: The right panel shows a single blue dot in the top 10 targets, yet the left panel does not have a blue dot.

We apologise for this source of confusion. This is because we only colored the genes that are above the threshold line (which is mainly for visualization purposes) in the left panel. To make this figure clearer, we synchronised the colors of the dots in the volcano plot with the ones of the top 10 targets (Figure 2).

Figure 2 - Differentially regulated transcription factor, kinase and phosphatase activities cancer vs.healthy tissue

A) Bar plot displaying the Normalised Enrichment Score (NES, proxy of activity change) of the 25 up or down regulated TF and top 25 up or down regulated kinase and phosphatases activities between kidney tumor and adjacent healthy tissue. B) Right panel shows the 10 most changing RNA abundances of the STAT2 regulated transcripts. Left panel shows the change of abundances of all STAT2 regulated transcripts that were used to estimate its activity change. X axis represents log fold change of regulated transcripts multiplied by the sign of regulation (-1 for inhibition and 1 for activation

of transcription). Y axis represents the significance of the log fold change ($-\log_{10}$ of p-value). C) Right panel shows the 10 most changing phospho-peptide abundances of the CDK7 regulated phospho-peptides. Left panel shows the change of abundances of all CDK7 regulated phospho-peptides that were used to estimate its activity change.

-Figure 4: Recommend black edges for visibility. Also, counter-intuitive that negative regulation/inhibition uses a pointed arrow while positive regulation/activation uses a flat arrow. Examination of the colors in the network and the arrows suggest that perhaps your legend is backwards for the Arrow Shape Effect?

Indeed, this was an error. The new network figure has fixed this. We have also changed the figure colours to use black edges only (Figure 4).

-Figure 4: For a kinase, can you make the distinction between increased activity as measured by differential expression as opposed to increased activity as measured by phosphosite enrichment? Similarly, does a transcription factor have increased activity because expression levels of the transcription factor itself are increased or because the targets of the transcription factor are increased?

In both cases, the activity estimation of a TF and kinase depends only on the abundance changes observed in its target transcript and phosphosites, respectively. The abundance changes measured in the transcript of phosphosite of a given transcript/kinase does not influence its activity estimation in any way.

As mentioned in our response to a previous comment (page 33), we now explain the concept of footprint based activity estimation more explicitly in the manuscript text by adding the following paragraph:

“Footprint-based activity estimation relies on the concept that measured omics abundances (such as phosphopeptides or transcripts) can be used as a proxy of up-stream (direct or indirect) regulator activities that are responsible for those changes (Casado et al, 2013; Ochoa et al, 2016; Rhodes et al, 2005). In the case of TF activity estimation, this translates into the fact that measured changes in the abundances of given transcripts give us information about the changes of activities of the transcription factors that regulates such abundances (Dugourd and Saez-Rodriguez 2019). This means that an activity estimation only depends on the changes of the abundances measured in its target transcripts, not its own transcript abundance. In this study, we use the VIPER algorithm (Alvarez et al. 2016) to estimate the activity of transcription factors and kinases based on transcript and phosphopeptide abundances changes, respectively.”

3. Discussion:

"It also predicted a depletion of adenine and consequently the down-regulation of PDPK1 activity through CXCR4." Adenine appears to be an input node in Figure 4A, so the decrease is observed not predicted?

We apologize for the confusion, we have written adenine but meant adenosine in the text. We have corrected this error.

Careful discussion of what is actually observed as opposed to what the model predicts is necessary.

For clarity we changed the network figures, the legend indicates that measured nodes (used as input and measurements in COSMOS) are marked with a star. Furthermore, we have made this distinction clear in the new paragraph that we wrote in our response to the second comment of Reviewer #1. For example we now describe COSMOS results as such :

“The COSMOS model suggests that MYC upregulates the expression of the metabolic enzyme BCAT1...”

4. General:

-By specifically referencing either COSMOS or CARNIVAL the authors imply that they are separate tools. It would be more useful if CARNIVAL were integrated into COSMOS for utility and also in the manuscript as switching between the two names throughout the manuscript is challenging.

We apologize for any confusion about the names. We addressed this point in our answer to your first general comment. As you suggest, CARNIVAL was replaced by COSMOS in the text whenever it was possible. In the newer parts, we also ensured that we referred to COSMOS more consistently. An example of such changes is given in our response to your first comment.

Additionally, it is unclear the output of COSMOS/CARNIVAL will be for future users.

The output of COSMOS is a set of two networks, ‘forward’ and ‘backward’, connecting two sets of input and measurement in both directions. We also provide tools to combine these two networks into one and to visualise subparts of the network easily. Thus, we made it a distinct R package to make it easier for users to know which to use in which context for non-familiar users.

It would be valuable to provide documentation on using the tool as part of the manuscript review.

We have now made COSMOS available as an R-package, with a complete explanation and tutorial at: <https://github.com/saezlab/COSMOS>

-Throughout the manuscript authors refer to transcription factor and kinase activity as measured by transcript expression. Activity is a poor word choice as it specifically refers to catalytic activity. The phosphosite data demonstrates an enrichment of a particular substrate phosphorylation event, not necessarily increased general kinase activity. Similarly, increased expression of genes targeted by a particular transcription factor does not mean that this particular transcription factor has increased activity - transcription factors do not have catalytic activity.

We acknowledge that the term of activity overlaps with its original meaning in the context of enzymatic activity from chemistry. At the same time, this is a commonly used term in the field of footprint analysis such as TF enrichment or Kinase enrichment analysis. Regarding phosphosites in particular, the ground assumption of kinase enrichment analysis is that the enrichment of the substrates’ phosphorylation of a given kinase can be used as a proxy of

the activity change of that kinase. We agree that this terminology, while broadly used, isn't perfect and we acknowledge this in the result section:

“By the term “activity”, we refer to a quantifiable proxy of the function of a protein, estimated based on the footprint left by said activity. This definition can apply, but is not limited to, an enzyme’s catalytic activity.”

-Throughout the manuscript gene names are not italicized.
We have modified the text accordingly.

-Throughout the manuscript numbers in the thousands do not have commas. ex. 32586 instead of 32,586
We have formatted the numbers accordingly.

-Throughout the manuscript there are inconsistencies in how tool names are reported. ex. limma vs. LIMMA, Omnipath vs. OmniPath, DOROTHEA vs. DoRotheA
We have modified the text accordingly.

1st Revision - Editorial Decision

14th Dec 2020

Thank you again for sending us your revised manuscript. We have now heard back from reviewer #1 who agreed to evaluate your study. As you will see below, this reviewer is satisfied with the modifications made and is supportive of publication.

Before we can formally accept the manuscript for publication we would ask you to address a few remaining editorial issues listed below.

REFEREE REPORTS

Reviewer #1:

We thank the authors for addressing our comments and concerns. We believe that now the manuscript is suitable for publication.

2nd Authors' Response to Reviewers

18th Dec 2020

The authors have made all requested editorial changes.

2nd Revision - Editorial Decision

21st Dec 2020

Thank you again for sending us your revised manuscript. We are now satisfied with the modifications made and I am pleased to inform you that your paper has been accepted for publication.

Corresponding Author Name: Pr. Julio Saez-rodriguez

Manuscript Number: MSB-20-9730